# Divergent folding-mediated epistasis among unstable membrane protein variants

Laura M Chamness[1], Charles P Kuntz[2], Andrew G McKee[1], Wesley D Penn[1], Christopher M Hemmerich[3], Douglas B Rusch[3], Hope Woods[4,5], Dyotima[1], Jens Meiler[4,6], Jonathan P Schlebach[2]*

[1]Department of Chemistry, Indiana University, Bloomington, United States; [2]The James Tarpo Jr. and Margaret Tarpo Department of Chemistry, Purdue University, West Lafayette, United States; [3]Center for Genomics and Bioinformatics, Indiana University, Bloomington, United States; [4]Department of Chemistry, Vanderbilt University, Nashville, United States; [5]Chemical and Physical Biology Program, Vanderbilt University, Nashville, United States; [6]Institute for Drug Discovery, Leipzig University, Leipzig, Germany

*For correspondence:
jschleba@purdue.edu

**Abstract** Many membrane proteins are prone to misfolding, which compromises their functional expression at the plasma membrane. This is particularly true for the mammalian gonadotropin-releasing hormone receptor GPCRs (GnRHR). We recently demonstrated that evolutionary GnRHR modifications appear to have coincided with adaptive changes in cotranslational folding efficiency. Though protein stability is known to shape evolution, it is unclear how cotranslational folding constraints modulate the synergistic, epistatic interactions between mutations. We therefore compared the pairwise interactions formed by mutations that disrupt the membrane topology (V276T) or tertiary structure (W107A) of GnRHR. Using deep mutational scanning, we evaluated how the plasma membrane expression of these variants is modified by hundreds of secondary mutations. An analysis of 251 mutants in three genetic backgrounds reveals that V276T and W107A form distinct epistatic interactions that depend on both the severity and the mechanism of destabilization. V276T forms predominantly negative epistatic interactions with destabilizing mutations in soluble loops. In contrast, W107A forms positive interactions with mutations in both loops and transmembrane domains that reflect the diminishing impacts of the destabilizing mutations in variants that are already unstable. These findings reveal how epistasis is remodeled by conformational defects in membrane proteins and in unstable proteins more generally.

## eLife assessment

This **important** study describes exhaustive deep mutational scanning (DMS) of the gonadotropin-releasing hormone wild-type receptor and for two single point mutations that impact its folding and structure, monitoring how plasma membrane expression levels are influenced by the introduced mutations. With **solid** evidence, the authors have pioneered an exploration of the interaction between mutations (epistasis) in a membrane protein, with a potential for explaining membrane protein evolution and genetic diseases.

**Figure 1.** Mutagenic perturbation of GnRHR folding. (**A**) A cartoon depicts the manner in which nascent polytopic membrane proteins are synthesized and folded at the endoplasmic reticulum (ER) membrane. The nascent protein first passes from a ribosome (gray) to a translocon (yellow), which facilitates its cotranslational membrane integration (stage I folding). Once the protein establishes its topology with respect to the membrane, it can fold into its native tertiary structure (stage II folding). (**B**) A homology model of *M. musculus* GnRHR (mGnRHR, green) is aligned with a crystal structure of *H. sapiens* GnRHR (hGnRHR, PDB 7BR3, gray). (**C**) A cross section of the mGnRHR model depicts the structural context of the V276 side chain (red). This side chain is located in transmembrane domain 6 and is exposed to the lipid bilayer. (**D**) A side view of the mGnRHR model depicts the structural context of the W107 side chain within extracellular loop 1. A hydrogen bond network between the W107 side chain indole, the backbone carbonyl oxygen of G110, and the backbone amide hydrogen from C114 is shown with dashes for reference.

The online version of this article includes the following figure supplement(s) for figure 1:

**Figure supplement 1.** Membrane integration of mGnRHR TM domain 6 and V276T.

## Introduction

Mutational effects on protein stability have important consequences for evolution. Destabilized proteins misfold more often, which can attenuate their ability to function in the cell (*Bloom et al., 2007*; *Marinko et al., 2019*). Though natural selection typically maximizes fitness by incorporating mutations that produce new or improved functions, this optimization process is often hindered by the destabilizing effects of most random mutations (*Bloom et al., 2007*; *Tokuriki et al., 2007*). The cumulative energetic effects of these mutations on protein stability are generally capable of creating non-additive, context-dependent epistatic interactions (*Gong et al., 2013*; *Olson et al., 2014*; *Starr and Thornton, 2016*). Nevertheless, there are many different aspects of protein synthesis, folding, and assembly that are governed by distinct energetic constraints. While epistatic interactions arising from changes in protein stability have been previously characterized in soluble proteins (*Gong et al., 2013*; *Olson et al., 2014*; *Faber et al., 2019*; *Nedrud et al., 2021*), the impact of epistasis on membrane protein folding and stability has yet to be explored. Although many of the mechanistic underpinnings of pairwise epistasis in soluble proteins are likely to be generalizable to all proteins, membrane proteins undergo additional cotranslational folding reactions that are governed by mechanistically distinct kinetic and energetic constraints (*Marinko et al., 2019*; *Schlebach and Sanders, 2015*). Based on these considerations, we suspect that mutations in membrane proteins could potentially modify cotranslational processes in a manner that generates distinct epistatic interactions that bias their evolutionary pathways in unique ways.

Unlike water-soluble proteins, membrane proteins must be cotranslationally inserted into lipid bilayers (stage I folding) in order to fold into their native structure (stage II folding, see *Figure 1A*; *Popot and Engelman, 1990*). In eukaryotes, stage I folding is primarily facilitated by Sec61 and a spectrum of various other translocon complexes (*Marinko et al., 2019*; *Smalinskaitè and Hegde, 2023*). Ribosomes translating membrane proteins are targeted to this translocon complex at the endoplasmic reticulum (ER) membrane, where the nascent transmembrane domains (TMDs) partition into the lipid bilayer and establish their native topology relative to the membrane (stage I) (*Pfeffer et al., 2016*; *Popot and Engelman, 1990*; *Van den Berg et al., 2004*). After translation is complete, membrane proteins fold and assemble (stage II) in a manner that is coupled to their passage through the secretory pathway to the plasma membrane and/or other destination organelles (*Marinko et al., 2019*; *Popot and Engelman, 1990*; *Wiseman et al., 2007*). Though they are governed by distinct physicochemical constraints, failures in either stage I or II folding are capable of increasing the proportion of nascent membrane proteins that are retained in the ER and prematurely degraded (*Marinko et al., 2019*; *Wiseman et al., 2007*). We therefore expect that epistatic interactions can arise from the cumulative energetic effects of mutations on either of these processes. Given that three-dimensional protein structures are stabilized by similar energetic principles in water and lipid bilayers, mutations that modify the fidelity of stage II folding energetics should potentially generate long-range epistatic interactions that are comparable to those observed in soluble proteins (*Marinko et al., 2019*). By comparison, the topological transitions that happen during the early stages of membrane protein synthesis may be coupled to neighboring loops and helices in a manner that can play a decisive role in the formation of the native topology (*Hessa et al., 2007*; *Ojemalm et al., 2012*; *White and von Heijne, 2008*). Based on these mechanistic distinctions and their association with different components of the proteostasis network, we reason that the effects of mutations on these two processes could potentially create complex and/or high-order epistatic interactions (*Morrison et al., 2021*; *Sailer and Harms, 2017*).

To gain insights into the mechanistic basis of epistatic interactions in membrane proteins, we surveyed the effects of single and double mutants on the plasma membrane expression (PME) of the gonadotropin-releasing hormone receptor (GnRHR), a G-protein coupled receptor (GPCR) involved in reproductive steroidogenesis across many species (*Janovick et al., 2013*; *Janovick et al., 2006*). We have previously demonstrated that various evolutionary sequence modifications within mammalian GnRHRs have tuned its fitness by compromising the fidelity of stage I folding (*Chamness et al., 2021*). Nevertheless, it remains unclear how the inefficient cotranslational folding of GnRHR reshapes its tolerance of secondary mutations. Here, we utilize deep mutational scanning (DMS) to carry out a focused analysis of 251 mutations in the background of two mouse GnRHR (mGnRHR) variants that selectively compromise either stage I or II folding. Our results show that mutations which generate stage I and II folding defects form distinct epistatic interactions throughout this receptor. An unsupervised learning analysis reveals how these interactions depend on both changes in stability and the topological context of the mutation. Together, these findings suggest that the distinct biosynthetic mechanisms of integral membrane proteins may differentially shape their fitness landscapes. The general implications of our findings for the role of protein folding and stability in protein evolution are discussed.

## Results
### Mutational library design

To compare how perturbations of stage I and II folding shape epistatic interactions, we first identified and characterized two individual mutations that are likely to selectively disrupt either the cotranslational membrane integration (V276T) or the native tertiary structure (W107A) of mGnRHR. We previously showed that T277 in TMD6 of human GnRHR (hGnRHR) contributes to the inefficient translocon-mediated membrane integration of TMD6, which decreases the receptor's PME (*Chamness et al., 2021*). A recent crystallographic structure of hGnRHR confirms that the polar side chain of T277, which corresponds to V276 in the mouse receptor (89.3% sequence identity), is exposed to the lipid bilayer and does not appear to make interhelical contacts that stabilize the native structure (*Figure 1B and C*; *Yan et al., 2020*). Thus, mutations at this position modify the hydrophobicity of TMD6 without perturbing the native tertiary contacts that stabilize the native fold. Indeed, replacing the native valine

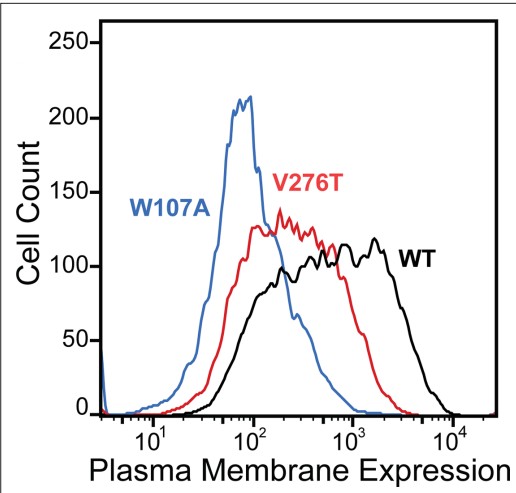

**Figure 2.** Plasma membrane expression of mGnRHR variants. Surface immunostaining and flow cytometry were used to compare the plasma membrane expression of select mGnRHR variants expressed in HEK293T cells. A histogram from one representative biological replicate depicts the distribution of plasma membrane expression intensities among cells expressing WT (black), V276T (red), or W107A (blue) mGnRHR.

residue in TMD6 of the mouse receptor with the threonine residue found in hGnRHR reduces the efficiency of its translocon-mediated membrane integration in vitro (*Figure 1—figure supplement 1*, *Supplementary file 1*). Furthermore, this substitution decreases the PME of the mouse receptor by 65 ± 4% relative to that of wild-type mGnRHR (WT, *Figure 2*). Consistent with previous findings, these results confirm that the inefficient translocation of this helix promotes cotranslational GnRHR misfolding (*Chamness et al., 2021*). In contrast, substitutions at W107 within extracellular loop 1 (ECL1) preserve the hydrophobicity of TMDs and their topological energetics while disrupting a conserved hydrogen bond network that is found in a wide array of GPCR structures (*Figure 1D*; *Jones et al., 2020*). Mutating this conserved tryptophan to alanine (W107A) reduces the PME of mGnRHR by 88 ± 4% relative to WT (*Figure 2*), which suggests that disrupting this hydrogen bond network destabilizes mGnRHR in a manner that promotes its cellular misfolding. Comparing how these mutations modify the proteostatic effects of secondary mutations will therefore reveal how changes in the fidelity of stage I (V276T) and stage II (W107A) folding differentially impact mutational epistasis.

To survey the epistatic interactions formed by these mutations, we generated a series of genetic libraries consisting of 1615 missense variants in the background of V276T, W107A, and WT mGnRHR. To ensure adequate dynamic range in the downstream assay, we created these variants in the cDNA of mGnRHR, which exhibits intermediate, tunable expression (*Chamness et al., 2021*). Briefly, we used a structural homology model of mGnRHR to select 85 residues distributed across the loops and helices of mGnRHR. We included all 19 amino acid substitutions at the 3 most solvent-accessible and the 3 most buried residues in each TMD (*Figure 3—figure supplement 1*, *Supplementary file 2*). We also included mutations encoding all 19 amino acid substitutions at positions that are evenly distributed across each soluble domain. We then generated a mixed array of mutagenic oligonucleotides that collectively encode this series of substitutions (*Supplementary file 3*) and used nicking mutagenesis to introduce these mutations into the V276T, W107A, and WT mGnRHR cDNAs (*Medina-Cucurella et al., 2019*), which produced three mixed plasmid pools. We generated each of these one-pot plasmid libraries in the context of vectors that contain a randomized ten-base region within the backbone. Unique ten-base plasmid identifiers that could be matched to a specific mGnRHR variant were used to score variants in the downstream assay using Illumina sequencing. Using whole plasmid PacBio sequencing of each library, we matched 1383 of the 1615 possible variants to one or more ten-base unique molecular identifier (UMI) in at least one of the three libraries, 320 of which were found in all three genetic backgrounds (see 'Materials and methods' for details).

## Plasma membrane expression of GnRHR mutant libraries

To identify secondary mutations that form epistatic interactions with V276T and W107A, we measured the relative PME of single and double mutants within each genetic library by DMS as previously described (*Penn et al., 2020*). Briefly, we first used the genetic libraries described above to generate three recombinant, pooled HEK293T cell lines in which the individual cells each express one of the single or double mGnRHR mutants of interest. A comparison of the mGnRHR surface immunostaining profiles of these cellular populations that recombinantly express this collection of variants in context of the V276T, W107A, or an otherwise WT genetic background reveals striking context-dependent

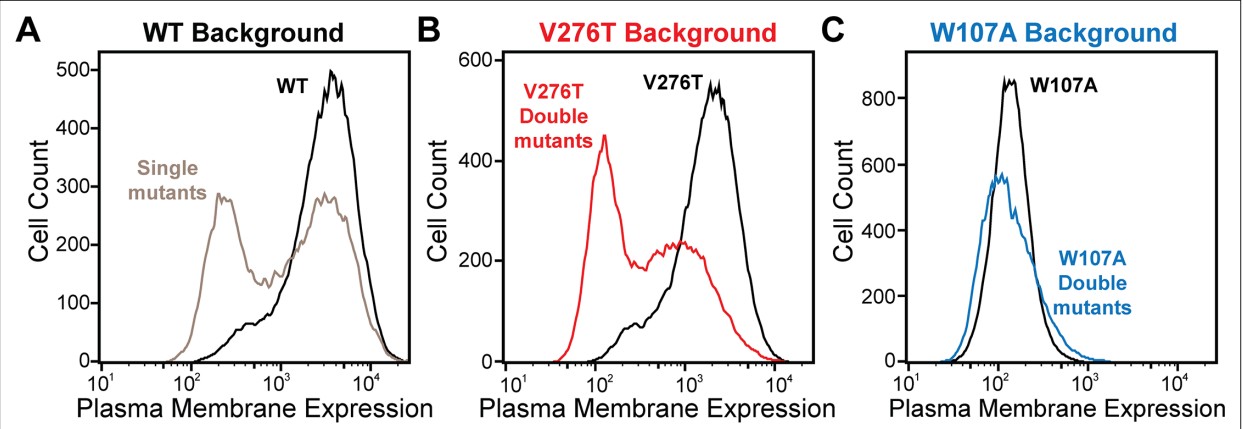

**Figure 3.** Plasma membrane expression of mGnRHR cellular libraries. mGnRHR mutant libraries, as well as their respective controls (WT or V276T/W107A single mutants, black) were expressed in mammalian cells, and the amount of mGnRHR expressed at the plasma membrane was measured during fluorescence-activated cell sorting. Histograms show the distributions of plasma membrane expression for mGnRHR mutants in the (**A**) WT library (gray), (**B**) V276T library (red), and (**C**) W107A library (blue).

The online version of this article includes the following figure supplement(s) for figure 3:

**Figure supplement 1.** Structural context of mGnRHR library mutants.

differences in their proteostatic effects. Recombinant cells expressing these variants in the WT background exhibit a bimodal distribution of mGnRHR surface immunostaining, where 59% of cells exhibit immunostaining that is comparable to WT and 41% of cells exhibit reduced expression relative to WT (*Figure 3A*). Although cells expressing this same collection of variants in the V276T background also exhibit bimodal surface immunostaining, 48% of the population exhibit diminished expression relative to the V276T variant (*Figure 3B*). This observation suggests many of these variants synergistically decrease mGnRHR expression in combination with the V276T mutation. Unlike the WT and V276T libraries, cells expressing this same collection of variants in combination with the W107A mutation exhibit a continuous range of surface immunostaining intensities that are only slightly lower, on average, relative to cells expressing the W107A single mutant (*Figure 3C*). The lack of dispersion likely reflects a compression of the distribution that arises as the folding efficiency of these variants approaches zero (see 'Discussion'). Cells expressing this library of W107A double mutants do, however, still contain cellular subpopulations with surface immunostaining intensities that are well above or below that of the W107A single mutant, which suggests that this fluorescence signal is sensitive enough to detect subtle differences in the PME of these variants (*Figure 3C*). Indeed, the mean fluorescence intensity associated with the surface immunostaining of the WT, V276T, and W107A mutant libraries were respectively 33.5-fold, 13.5-fold, and 2.6-fold higher than the background, on average, across the two biological replicates. Overall, these results suggest that none of the secondary mutations can fully compensate for the proteostatic effects of the W107A mutation.

To compare the PME of these variants in each genetic background, we fractionated each of the three mixed cell populations into quartiles based on the relative surface immunostaining of their expressed mGnRHR variants using fluorescence-activated cell sorting (FACS), then extracted the genomic DNA from each cellular fraction. We then used Illumina sequencing to track the relative abundance of the recombined UMIs and their associated variants within each fraction and used these measurements to estimate the corresponding PME of each variant. A series of filters relating to the quality of the reads, the sampling of each mutation, and the similarity of variant scores across replicates were used to remove poorly sampled variants with unreliable scores (see 'Materials and methods'). Briefly, to restrict our analysis to variants that can be reliably scored, we removed variants that did not have at least 50 counts across the four bins in each replicate. Additionally, we compared the percentile rank of each variant across replicates and discarded variant scores for which the difference in percentile rank across the two replicates was greater than 25%. The relative PME of the remaining variants was averaged across two biological replicates. Overall, we measured the relative PME of 1004 missense variants in the WT background, 338 missense variants in the V276T background, and 1179 missense variants in the W107A background (*Supplementary file 4*). Out of the 320 variants identified in all

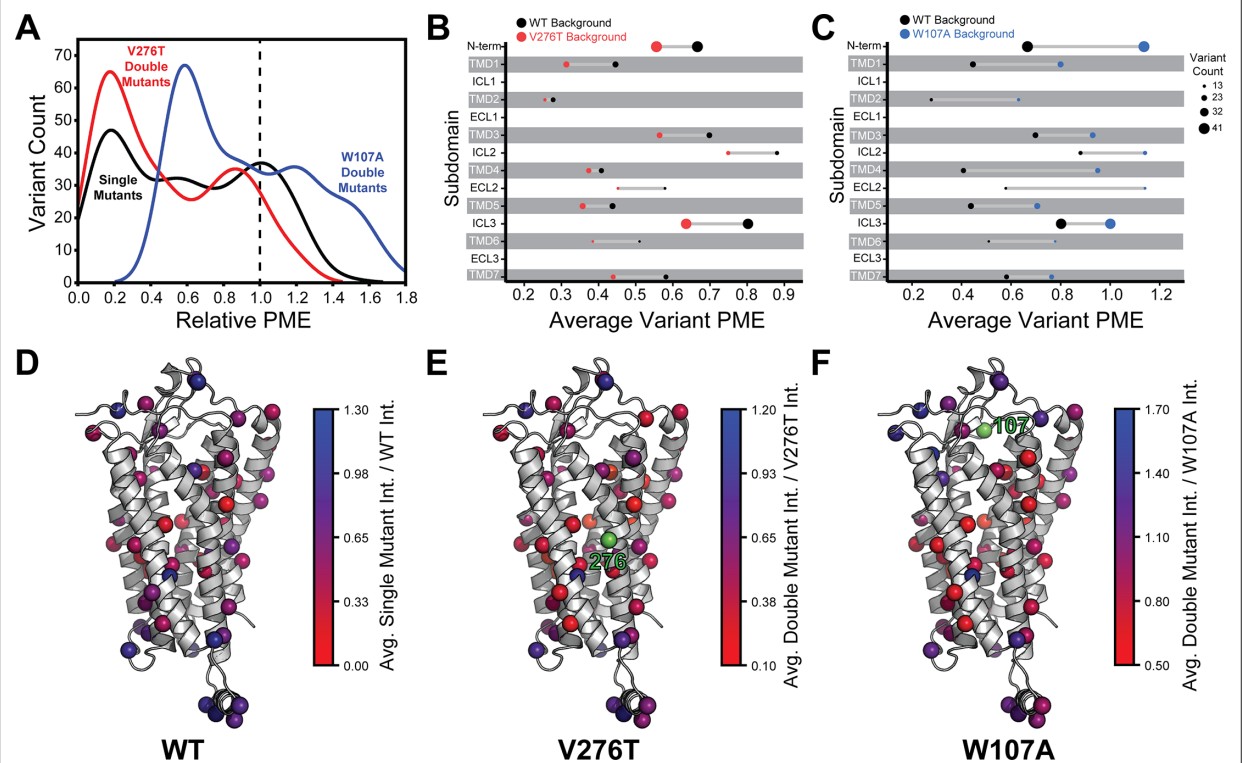

**Figure 4.** Relative plasma membrane expression of mGnRHR mutants. (**A**) A histogram shows the distribution of plasma membrane expression measurements relative to their respective controls (WT or V276T/W107A single mutants) for the WT (black), V276T (red), and W107A (blue) mutational libraries. A dashed line at the normalized plasma membrane expression value of 1.0, representing no mutational effect relative to the control, is displayed for reference. (**B, C**) The average relative intensities across each domain are compared between the V276T (**B**, red) and W107A (**C**, blue) double mutant libraries and the single mutant library (black). The size of each dot represents the number of variants measured. (**D–F**) Average relative intensities for the (**D**) WT library, (**E**) V276T library, and (**F**) W107A library were projected onto a homology model of mGnRHR. Scores represent the average value from two biological replicates. Residues 276 and 107 are displayed in green for reference. Individual variant scores can be found in *Supplementary file 4*.

The online version of this article includes the following figure supplement(s) for figure 4:

**Figure supplement 1.** Correlation of deep mutational scanning (DMS) variant intensity values across two biological replicates.

three plasmid libraries by PacBio sequencing, 251 mutations passed these quality filters in the Illumina sequencing of the cellular isolates (*Supplementary file 4*). The intensity values for these variants were highly correlated across independent biological replicates, which demonstrates that these measurements exhibit precision despite variations in the signal-to-noise associated with their surface immunostaining (*Figure 4—figure supplement 1*). In the following, we will focus on the comparison of the effects of these 251 mutations in each background in order to determine how their proteostatic effects are modified by secondary mutations.

To facilitate the comparison of estimated immunostaining intensities for each variant across the three genetic backgrounds, we normalized the raw immunostaining intensity values for each variant relative to that of the measured intensity values for the corresponding reference sequence (V276T, W107A, or WT mGnRHR). Relative intensity values of 1.0 correspond to no effect, whereas values over 1.0 correspond to mutations that enhance surface expression in that genetic background, and vice versa. A histogram of the relative intensity values for the 251 variants measured in all three backgrounds generally recapitulates the trends in the cellular immunostaining histograms (*Figures 3 and 4A*). Most of these mutations decrease the PME of mGnRHR in all three backgrounds, which reflects the limited mutational tolerance of membrane proteins (*Figure 4A*; *Telenti et al., 2016*). Notably, only a modest fraction of mutations measurably enhance the surface expression of WT mGnRHR (51 mutations, 20%) and V276T mGnRHR (21 mutations, 8% *Figure 4A*). Mutations in the V276T background tend to decrease surface expression more than they do in the WT background, and this trend

persists across most domains of the protein (*Figure 4A and B*). In contrast, a larger proportion of these mutations (100 mutants, 40%) enhance the surface expression of W107A (*Figure 4A*), though these increases in immunostaining are relatively modest (*Figure 3C*). Furthermore, the mutations tend to be better tolerated with respect to surface expression in combination with W107A relative to their effects on expression in the WT background (*Figure 4C*).

Projecting the average relative intensities for the mutations at each residue onto the structure reveals that most positions exhibit similar trends in the mutational tolerance in each background, with loop residues being generally more permissive than those within TMDs (*Figure 4D–F*). Nevertheless, there are subdomains where the quantitative mutagenic effects deviate in a context-dependent manner. Variants bearing mutations within the C-terminal regions including ICL3, TMD6, and TMD7 fare consistently worse in the V276T background relative to WT (paired Wilcoxon signed rank test p-values of 0.0001, 0.02, and 0.005, respectively) (*Figure 4B and E*). Given that V276T perturbs the cotranslational membrane integration of TMD6 (*Figure 1—figure supplement 1*, *Supplementary file 1*), this directional bias potentially suggests that the apparent interactions between these mutations manifest during the late stages of cotranslational folding. In contrast, mutations that are better tolerated in the context of W107A mGnRHR are located throughout the structure but are particularly abundant among residues in the middle of the primary structure that form ICL2, TMD4, and ECL2 (paired Wilcoxon signed rank test p-values of 0.0005, 0.0001, and 0.004, respectively) (*Figure 4C and F*). Together, these observations show how the proteostatic effects of mGnRHR mutations are modified by secondary mutations that differentially affect the fidelities of stage I and II folding.

## Distinct pairwise epistasis in V276T and W107A GnRHR

To compare epistatic trends in these libraries, we calculated epistasis scores (ε) for the interactions that these 251 mutations form with V276T and W107A by comparing their relative effects on PME of the WT, V276T, and W107A variants using a previously described epistasis model (product model, see 'Materials and methods'; *Olson et al., 2014*). Positive ε values denote double mutants that have greater PME than would be expected based on the effects of single mutants. Negative ε values denote double mutants that have lower PME than would be expected based on the effects of single mutants. Pairs of mutations with ε values near zero have additive effects on PME. For most double mutants, epistasis scores are near zero, suggesting that many of the 251 random mutations have additive effects on PME in each background. This result is consistent with evidence that epistatic interactions are typically rare in the context of stable proteins (*Starr and Thornton, 2016*). Nevertheless, the distributions of epistasis scores for the two double mutant libraries are shifted in opposite directions. The mutations generally form positive epistatic interactions with W107A and negative epistatic interactions with V276T (*Figure 5*). Interestingly, the difference in the epistasis scores for the interactions these variants form with W107A and V276T was at least 1.0 for 98 of the 251 variants (*Supplementary file 4*). Notably, these 98 variants are enriched with TMD variants (65% TMD) relative to the overall set of 251 variants (45% TMD, Fisher's exact test p=0.0019). These findings suggest random mutations form epistatic interactions in the context of unstable mGnRHR variants in a manner that depends on the specific folding defect (V276T vs. W107A) and topological context.

## Molecular basis for the observed epistatic interactions

To identify general trends in the observed epistatic interactions, we utilized unsupervised learning to elucidate patterns among the relative PME values of variants across each genetic background. We first utilized uniform manifold approximation projection (UMAP) (*McInnes et al., 2018*) to identify mutations that have similar expression profiles across these conditions. A projection of the

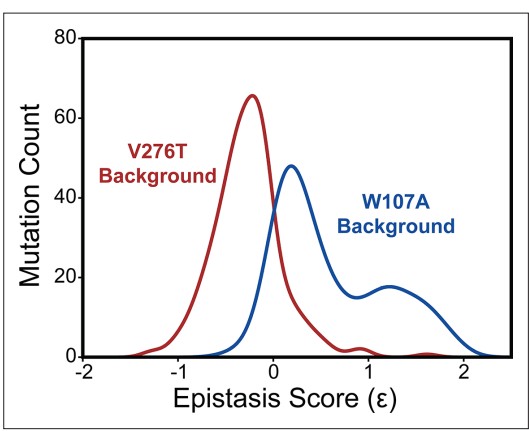

**Figure 5.** Epistasis in the mGnRHR double mutant libraries. A histogram depicts the distribution of epistasis scores (ε) for interactions the subset of 251 high-quality secondary mutations form with V276T (red) and W107A (blue).

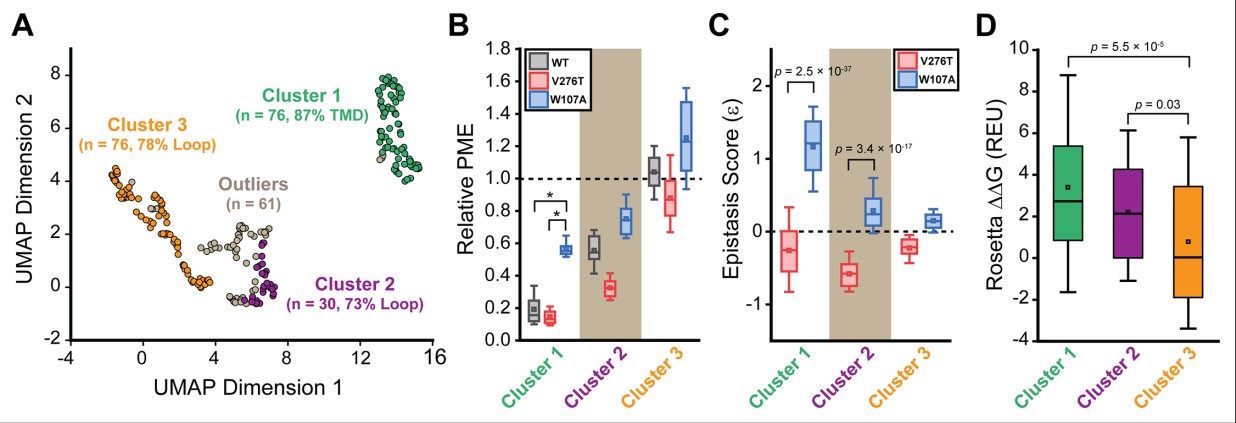

**Figure 6.** General trends in mGnRHR epistasis. Trends associated with the observed pairwise epistasis within mGnRHR are identified using unsupervised learning. (**A**) Uniform manifold approximation projection (UMAP) was used to differentiate variants based on differences in their relative expression in the V276T, W107A, or WT background. Variants are projected onto an arbitrary two-dimensional coordinate based on the results and are colored according to whether they were assigned to cluster 1 (green), cluster 2 (purple), cluster 3 (orange), or were designated as outliers (gray) by HDBSCAN. The percentage of the mutations that fall within transmembrane domains (TMDs) or loops are shown for reference. (**B**) A box and whisker plot depicts the statistical distributions of relative plasma membrane expression (PME) values among variants within each cluster in the context of V276T (red), W107A (blue), or WT (gray) mGnRHR. Select clusters of variants that exhibit statistically different expression profiles according to a Mann–Whitney U-test are indicated (*p<0.001). A value of 1 corresponds to mutations that have no effect on the PME of mGnRHR in the indicated genetic background. (**C**) A box and whisker plot depicts the statistical distribution of epistasis scores associated with the interactions between the mutations within each cluster and either V276T (red) or W107A (blue). p-values for select Mann–Whitney U-tests comparing the interactions of these mutations with V276T and W107A are indicated. A value of 0 indicates that the effects of the two mutations are additive. (**D**) A box and whisker plot depicts the statistical distribution of Rosetta ΔΔG values among mutations within each cluster. p-values for select Mann–Whitney U-tests comparing the ΔΔG values across clusters are shown for reference. For panels (**B–D**), the edges of the boxes correspond to the 75th and 25th percentile values while the whiskers reflect the values of the 90th and 10th percentile. The central hash and square within the box represent the average and median values, respectively. These analyses were carried out on a subset of 243 variants with high-quality expression measurements with calculable Rosetta ΔΔG values.

variants onto a resulting two-dimensional coordinate reveals that most variants fall into one of approximately three groups (*Figure 6A*). Using a density-based hierarchical clustering analysis (HDBSCAN), we unambiguously assigned 182 variants into three distinct clusters based on their expression profiles alone (*Figure 6A*, *Supplementary file 4*; *Campello et al., 2013*). An analysis of the structural context of these mutants reveals that one of the largest clusters (cluster 3, n = 76) primarily consists of mutations of soluble loop residues that have little impact on expression and exhibit minimal epistasis (*Figure 6A–C*). This analysis also identified a smaller cluster of loop mutations (cluster 2, n = 30) that moderately decrease expression and exhibit a greater propensity to form epistatic interactions with V276T and/ or W107A relative to the neutral loop mutations in cluster 3 (*Figure 6A–C*). Nevertheless, most of the mutations that exhibit strong positive epistatic interactions with W107A fell into cluster 1 (n = 76), which primarily consists of mutations within TMDs that significantly decrease expression in all three genetic backgrounds (*Figure 6A–C*). A comparison across these clusters suggests positive epistatic interactions with W107A primarily stems from the fact that mutations that compromise expression can only cause a relatively modest reduction in the PME of W107A GnRHR (*Figure 6B and C*), which is already predominantly misfolded. In contrast, the moderately disruptive loop mutations in cluster 2 tend to exhibit more pronounced negative epistatic interactions with V276T than do the highly disruptive TMD mutations in cluster 1 (*Figure 6C*). The divergent epistatic interactions that disruptive loop and TMD mutations form with V276T potentially arise from differences in the mechanistic basis for the destabilization caused by these two classes of mutations (see 'Discussion').

Many epistatic interactions arise from the additive energetic effects of mutations on protein folding (*Olson et al., 2014*; *Tokuriki and Tawfik, 2009*). To determine how these trends relate to the effects of mutations on thermodynamic stability, we utilized Rosetta CM to generate structural models of 243 of the 251 variants that were scored in all three backgrounds and fall within the structured regions of the receptor (*Song et al., 2013*). We then utilized a membrane-optimized Rosetta energy scoring function to approximate the change in the energy of the native structural ensemble of each variant (*Alford et al., 2015*). The cluster of loop mutations with lower expression (cluster 2,

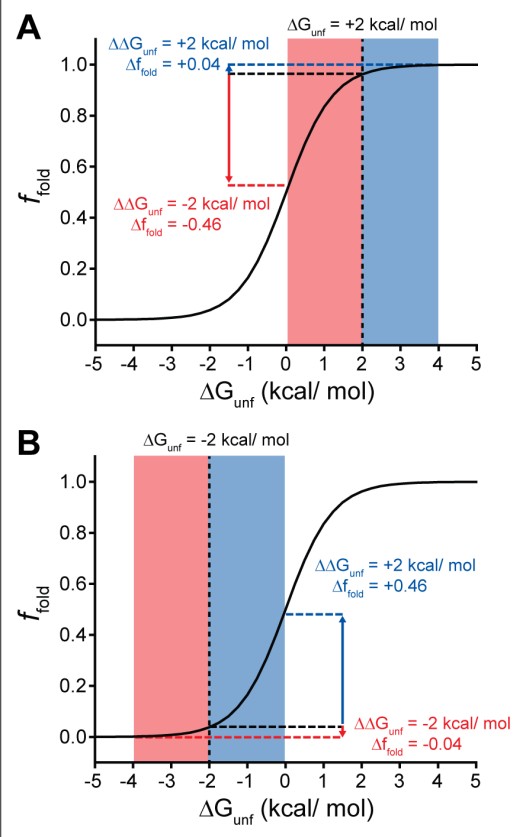

**Figure 7.** Context-dependent impacts of stabilizing and destabilizing mutations. Line plots depict the relationship between the equilibrium fraction of folded protein ($f_{fold}$) and the free energy of unfolding ($\Delta G_{unf}$). (**A**) A moderately stable protein ($\Delta G_{unf}$ = +2 kcal/mol) exhibits a greater change in $f_{fold}$ after acquiring one or more destabilizing mutations (total $\Delta\Delta G_{unf}$ = –2 kcal/mol, red) than after acquiring one or more mutations that stabilize the protein to the same extent (total $\Delta\Delta G_{unf}$ = +2 kcal/mol, blue). (**B**) A moderately unstable protein ($\Delta G_{unf}$ = –2 kcal/mol) exhibits a greater change in $f_{fold}$ after acquiring one or more stabilizing mutations (total $\Delta\Delta G_{unf}$ = +2 kcal/mol, blue) than after acquiring one or more mutations that destabilize the protein to the same extent (total $\Delta\Delta G_{unf}$ = –2 kcal/mol, red).

avg. $\Delta\Delta G$ = +2.2 REU) generally contains more destabilizing mutations relative to the cluster of loop mutations with WT-like expression (cluster 3, avg. $\Delta\Delta G$ = +0.8 REU), which affirms the general relationship between stability and PME in this system (p=0.003, *Figure 6B and D*). However, the cluster of TMD variants (cluster 1, avg. $\Delta\Delta G$ = +3.4 REU) is generally more destabilizing than either of the loop clusters (*Figure 6D*). A comparison across the clusters suggests that the degree of destabilization generally tracks with the magnitude of positive epistatic interactions that form with W107A, regardless of structural context (*Figure 6C and D*). This uptick in positive epistasis among mutations that destabilize the native fold and severely reduce PME likely reflects the attenuated effect of destabilizing mutations as the equilibrium fraction of folded protein approaches zero (see 'Discussion', *Figure 7*). In contrast to W107A, epistatic interactions with V276T are predominantly negative and are most pronounced among moderately destabilizing loop mutations (*Figure 6C and D*). The abundance of negative interactions with V276T parallels the observed pairwise epistasis in other folded proteins (*Olson et al., 2014*), and likely reflects the higher baseline expression and/or stability of this variant relative to W107A (*Figures 2 and 7*). Nevertheless, stability-mediated epistasis cannot account for the attenuation of the epistatic interactions that V276T forms with the highly destabilizing mutations within cluster 1 relative to the moderately destabilizing mutations within cluster 2 (*Figure 6C and D*). These observations suggest that the magnitude of the epistatic interactions formed by V276T is dependent on not only the change in stability, but also the topological context of the secondary mutation.

## Discussion

Many epistatic interactions between mutations within genes encoding soluble proteins arise from their cumulative effects on protein folding energetics (*Nedrud et al., 2021*; *Olson et al., 2014*). Folded proteins can only tolerate a limited number of destabilizing mutations before their cumulative energetic effects cause a cooperative decrease in the fraction of folded protein (*Figure 7A*). Although this general principle should apply to all proteins, it is unclear how the distinct energetic processes involved in membrane protein biosynthesis and folding may alter such evolutionary couplings. In this study, we utilize DMS to compare the epistatic interactions formed by two mutations that reduce the expression of a misfolding-prone GPCR by selectively destabilizing either its cotranslational membrane integration (V276T) or its native tertiary structure (W107A). We note that the introduction of the V276T mutation into the mouse receptor utilized herein mimics one of the key destabilizing modifications that reduces the expression of hGnRHR relative to mGnRHR (*Chamness et al., 2021*). Though the quantitative energetic effects of these mutations on cotranslational and/or post-translational folding processes remain unclear in the absence of a suitable biophysical assay, they each markedly reduce the plasma membrane expression of mGnRHR and it

seems highly unlikely that they should elicit identical quantitative effects on folding and assembly (*Figure 2*). Thus, a comparison of the epistatic interactions formed by these two mutations should reveal the extent to which the mechanism of misfolding impacts the propensity for genetic interactions between mutations. Through a focused analysis of 753 total single and double mutants, we find that these mutations give rise to divergent epistatic interactions that modify the PME of mGnRHR (*Figure 5*). Mutations that form epistatic interactions with V276T tend to synergistically decrease mGnRHR expression (*Figures 3A and B, 4B, and 5*), a typical epistatic trend that likely arises from the combined destabilizing effects of both mutations. Surprisingly, these interactions are most prominent among moderately destabilizing loop mutations (*Figure 6C*). In contrast, many of these same secondary mutations instead form positive epistatic interactions with W107A mGnRHR, regardless of their topological context (*Figures 5 and 6C*). As this comparison reflects the epistatic interactions formed by an identical set of secondary mutations, we conclude that these context-dependent epistatic interactions arise from differences in the magnitude and/or the mechanism of the mGnRHR destabilization caused by the V276T and W107A mutations. These observed trends potentially reflect a previously undescribed manifestation of 'ensemble epistasis' (*Morrison et al., 2021*) in which genetic interactions arise from the differential effects of mutations on the accumulation of distinct misfolded conformations that vary with respect to their recognition and/or degradation by cellular quality control pathways.

In the context of stable proteins, stability-mediated epistasis tends to magnify the effects of deleterious mutations due to the cumulative destabilization that arises from sequential random substitutions (*Tokuriki et al., 2007*). This is a fundamental consequence of the cooperative dependence of the fraction of folded protein on the free energy of unfolding. Indeed, most of the random mGnRHR mutations characterized herein are predicted to destabilize the native structure in a manner that coincides with a measurable decrease in the PME in the context of all three genetic backgrounds (*Figures 4A and 6D*). Whereas these destabilizing mutations exhibit the expected negative epistatic couplings with V276T, their interactions with W107A skew positive (*Figure 5*). This anomalous positive epistasis is potentially a by-product of the severe destabilizing effects of W107A relative to V276T (*Figure 2*). While destabilizing secondary mutations can cause significant decreases in the yield of folded WT or V276T mGnRHR, the W107A variant already exhibits marginal surface immunostaining. In the context of this predominantly misfolded, poorly expressed variant, even highly destabilizing secondary mutations can only cause a relatively small decrease in the yield of folded protein. This observation can be more generally related to the relatively shallow dependence of the fraction of folded protein on the free energy of folding under conditions where folding becomes unfavorable – unfolding transitions become shallow as the fraction of folded protein approaches zero (*Figure 7B*). Stabilizing mutations can also generate potent positive epistatic couplings in this context given that even small increases in stability can push the system into the steepest portion of the transition zone (*Figure 7B*). Paradoxically, the asymmetry of the folding curve in this regime suggests destabilizing mutations are more tolerable in the context of an unstable protein – the effects of these variants exhibit 'diminishing returns' (*Otto and Feldman, 1997*) because folding efficiency effectively has nowhere to go but upwards. Essentially, the nature of this sigmoidal transition suggests the typical trends associated with stability-mediated epistasis should become inverted as stability decreases. Such considerations could potentially expand the accessible sequence space in metastable proteins, which constitute a significant portion of the proteome (*Ghosh and Dill, 2010*; *Zeldovich et al., 2007*). Combinations of mutations that would not generally be tolerated in stable proteins may also occur through such mechanisms in the context of the numerous misfolded secondary alleles that are carried within the genomes of humans and other diploids with no appreciable fitness cost (e.g., the ΔF508 variant of *CFTR*).

Unlike W107A, the epistatic interactions formed by V276T appear to depend on their topological context- negative epistasis is more prevalent among moderately destabilizing loop variants than highly destabilizing TMD variants (*Figure 6C and D*). This observation suggests stability-mediated epistasis between mutations that destabilize the native tertiary structure of integral membrane proteins (stage II folding) may be distinct from the interactions that arise from the disruption of their translocon-mediated cotranslational membrane integration (stage I folding, *Figure 1A*). This divergence could potentially reflect the distinction between the energetic principles of protein folding relative to those that guide the insertion of proteins into membranes, which is fundamentally driven by a membrane depth-dependent solvation energetics (*Hessa et al., 2007*; *Moon and Fleming, 2011*).

Counterintuitively, polar mutations that disrupt the membrane integration of individual TMDs also have the potential to form interhelical hydrogen bonds that can stabilize the topological orientation of neighboring TMDs, which can give rise to long-range topological couplings. (*Hermansson and von Heijne, 2003*; *Meindl-Beinker et al., 2006*; *Ojemalm et al., 2012*). The propensity of such contacts to form may in some cases depend on the length and hydrophobicity of the loops that connect them (*Woodall et al., 2017*). Finally, we should note that complex epistatic interactions between residues within TMDs could potentially arise through their propensity to remodel the translocon complex and/or other quality control interactions – polar residues within TMDs may affect the recruitment of secondary insertases and/or intramembrane chaperones such as TRAP, the PAT complex, the BOS complex, the GEL complex, and the ER membrane protein complex (*Chen et al., 2023*; *Gemmer et al., 2023*; *Matreyek et al., 2017*; *Pleiner et al., 2020*; *Shurtleff et al., 2018*; *Smalinskaitẻ et al., 2022*; *Sundaram et al., 2022*). Additional insights into the mechanistic basis for the recruitment of such complexes may therefore be needed to fully rationalize evolutionary couplings between TMDs. We note that, more generally, we suspect that synergistic genetic interactions between mutations could potentially arise from changes in the energetics of many other biosynthetic processes such as cofactor binding, oligomer formation, and chaperone interactions (*Bershtein et al., 2013*; *Rodrigues et al., 2016*).

Together, our results reveal divergent, mechanism-dependent patterns of pairwise epistasis in the context of a misfolding-prone integral membrane protein. These findings suggest the folding-mediated epistasis is likely to vary among different classes of destabilizing mutations in a manner that should also depend on folding efficiency and/or the mechanism(s) of misfolding in the cell. These findings provide novel insights as to how the unique biosynthetic constraints of integral membrane proteins can potentially lead to unique evolutionary patterns. Moreover, our results generally suggest that the deleterious effects of destabilizing mutations are partially attenuated in the context of unstable proteins. Additional investigations are needed to explore how stability-mediated epistasis is modified by other proteostatic constraints, such as the interactions of nascent proteins with cofactors and/or molecular chaperones.

# Materials and methods

## Key resources table

| Reagent type (species) or resource | Designation | Source or reference | Identifiers | Additional information |
|---|---|---|---|---|
| Gene (*Mus musculus*) | GnRHR | GenBank | L01119 | - |
| Gene (*Homo sapiens*) | GnRHR | GenBank | L03380 | - |
| Strain, strain background (*Escherichia coli*) | NEB 10-β | New England Biolabs | C3020K | Electrocompetent |
| Cell line (*H. sapiens*) | HEK293T | ATCC | CRL-3216 | - |
| Transfected construct (*H. sapiens*) | Tet-Bxb1-BFP HEK293T | Laboratory of Doug Fowler | - | Clone 37 described in *Jones et al., 2020* |
| Biological sample (*Canis familiarus*) | Pancreatic rough microsomes | tRNA probes | - | - |
| Biological sample (*Oryctolagus cuniculus*) | Reticulocyte lysate | Promega | L4960 | Nuclease-treated |
| Antibody | Anti-Hemagglutinin Antibody (mouse monoclonal) | Invitrogen | 26183-D550 | 2–2.2.14, DyLight 550 conjugate |
| Recombinant DNA reagent | pGEM- Lep- TMD6 | *Hessa et al., 2007* | - | - |
| Recombinant DNA reagent | pcDNA5 CMV-HA-mGnRHR-IRES-eGFP | *Chamness et al., 2021* | - | - |
| Recombinant DNA reagent | pcDNA5 attB-HA-mGnRHR-IRES-eGFP | This paper | - | A plasmid described in *Chamness et al., 2021* was further modified as is described in 'Materials and methods' |

*Continued on next page*

*Continued*

| Reagent type (species) or resource | Designation | Source or reference | Identifiers | Additional information |
|---|---|---|---|---|
| Sequence-based reagent | Custom mGnRHR primer pool | Agilent Technologies Inc | - | See *Supplementary file 3* for sequences |
| Sequence-based reagent | Custom 10- base randomization primer | Integrated DNA Technologies | - | gcatgaagaatctgcttagggttaggcg nnnnnnnnnncttcgcgatgtacg ggccagat |
| Peptide, recombinant protein | PrimeSTAR HS DNA Polymerase | Takara Bio Inc | R010B | - |
| Peptide, recombinant protein | KAPA HiFi HotStart Ready Mix | Roche Diagnostics | 07958927001 | - |
| Commercial assay or kit | InFusion HD Cloning | Takara Bio Inc | 638944 | - |
| Commercial assay or kit | GenElute Mammalian gDNA Miniprep Kit | Sigma-Aldrich | G1N70 | - |
| Software, algorithm | Flowjo X | Treestar | - | - |
| Software, algorithm | OriginLab 2023 | OriginLab | - | - |

## Plasmid preparation and characterization of genetic libraries

The expression of transiently expressed single mutants was carried out using a previously described pcDNA5 FRT expression vector containing mGnRHR cDNA with an N-terminal hemagglutinin (HA) epitope and bicistronic eGFP (*Figure 2*; *Chamness et al., 2021*). Mutations were introduced by site-directed mutagenesis with PrimeSTAR HS DNA Polymerase (Takara Bio, Shiga, Japan). Biochemical measurements of the membrane integration of TMD6 were carried out using a previously described pGEM vector containing modified leader peptidase (Lep), with TMDs of interest cloned into the H-segment (*Figure 1—figure supplement 1*; *Chamness et al., 2021*; *Hessa et al., 2007*). Mutations were introduced into the H-segment by site-directed mutagenesis with PrimeSTAR HS DNA Polymerase (Takara Bio). Plasmids were purified using a ZymoPURE Midiprep Kit (Zymo Research, Irvine, CA). DMS experiments were carried out using a previously described pcDNA5 FRT vector containing mGnRHR cDNA bearing an N-terminal influenza hemagglutinin (HA) epitope, followed by an internal ribosome entry site (IRES) and eGFP sequence, which was further modified to be compatible with recombination-based DMS approaches (*Chamness et al., 2021*). First, an attB recombination site was inserted in place of the CMV promoter by InFusion HD Cloning (Takara Bio). To facilitate the creation of these libraries using nicking mutagenesis, a BbvCI restriction site was introduced by site-directed mutagenesis using PrimeSTAR HS DNA Polymerase (Takara Bio). The V276T and W107A mutations were also introduced by site-directed mutagenesis with PrimeSTAR HS DNA Polymerase (Takara Bio).

To generate mutational libraries, a 10N UMI, or 'barcode', was first inserted into the plasmid backbone using a previously described nicking mutagenesis method (*Wrenbeck et al., 2016*). Nicking mutagenesis plasmid products were transformed into NEB 10-beta electrocompetent cells (New England Biolabs, Ipswich, MA) and purified using a ZymoPURE Midiprep Kit (Zymo Research). A programmed oligo pool (Agilent, Santa Clara, CA) encoding all 19 amino acid substitutions at 85 residues throughout the mGnRHR sequence was then used to generate pools of single and double mutants in the context of WT, V276T, and W107A mGnRHR cDNA using a previously described nicking mutagenesis method (*Medina-Cucurella et al., 2019*). The resulting plasmid products were transformed into NEB 10-beta electrocompetent cells (New England Biolabs) and purified using a Zymo-PURE Midiprep Kit (Zymo Research). To limit the number of mGnRHR variants per UMI, these libraries were then bottlenecked through an additional transformation in NEB Turbo electrocompetent cells (New England Biolabs) and were again purified using a ZymoPURE Midiprep Kit (Zymo Research). The resulting plasmid preparations contained detectable levels of a concatemer product, which were removed from the mutational libraries by gel purification using a Zymoclean Gel DNA Recovery Kit (Zymo Research). Purified products were again transformed into either NEB 10-beta (New England Biolabs) or SIG10 (Sigma-Aldrich, St. Louis, MO) electrocompetent cells, and purified with a Zymo-PURE Midiprep Kit (Zymo Research).

To associate each UMI with its corresponding GnRHR variant, the final plasmid libraries were also sequenced using PacBio SMRT sequencing. Briefly, plasmid libraries were first double-digested with PmlI and MfeI-HF restriction enzymes (New England Biolabs), then purified using a Zymo DNA Clean & Concentrator-5 kit (Zymo Research). Complete digestion was verified by analyzing the products on

an agarose gel, purity was assessed with a Synergy Neo2 microplate reader (BioTek, Winooski, VT), and the final yield was quantified using a Qubit4 fluorometer (Thermo Fisher, Waltham, MA). The 1537 bp fragment for each library, containing the 10N UMI and GnRHR open reading frame, was isolated on a BluePippin instrument (Sage Science, Beverly, MA). These fragments were then assembled into PacBio libraries and sequenced on the Sequel II system with a 30 hr runtime.

The sequence of each plasmid-derived fragment containing mGnRHR variants and their corresponding UMIs was determined by reconstructing the circular consensus sequences (CCS) for each independent well in the PacBio readout. mGnRHR libraries were sequenced to a depth of 1,825,010 wells for WT, 1,545,151 wells for V276T, and 1,497,359 wells for W107A. A minimum coverage of five passes around each circular fragment (subread) was required for incorporation of sequencing data from an individual well into the analysis. CCS were mapped to the reference open reading frame using minimap2. CCS that did not contain a complete 10-base UMI sequence or fully cover the target ORF were excluded from the analysis. Variations in the mGnRHR sequence of individual fragments were called using SAMtools. Reads were then grouped according to their UMI sequences. Given that insertion and deletion (indel) artifacts are prevalent within PacBio sequencing data, we only removed UMIs from the analysis if indels were detected at a specific position within the UMI in at least 10% of the subreads. Indels observed within the open-reading frame were only called if their occurrence was judged to be statistically significant relative to the background indel rate for the reads according to a binomial test ($p \leq 0.05$). Codon substitutions encoded in our primer library (*Supplementary file 3*) were called within the mGnRHR open-reading frame only if they were detected in at least 75% of the individual subreads. Only UMIs that were found to be associated with a single mGnRHR variant were included in the final analysis. Our final 'dictionaries' for the WT, V276T, and W107A libraries contained 3,261, 531, and 4962 unique UMIs, respectively, that could be indexed to a corresponding mGnRHR variant.

## Preparation of cellular libraries and cell sorting

A previously described HEK293T cell line with a genomic Tet-Bxb1-BFP landing pad was obtained from D. Fowler (*Matreyek et al., 2017*). Short tandem repeat profiling was carried out to confirm that these cells are derived from HEK293T cells and PCR-based mycoplasma testing was carried out to ensure cultures were not contaminated (American Type Culture Collection, Manassas, VA). Cells were grown at 37°C and 5.0% $CO_2$ in complete media, made of Dulbecco's Modified Eagle's medium (DMEM, Gibco, Grand Island, NY) supplemented with 10% fetal bovine serum (FBS, Gibco), 0.5% penicillin (Gibco), and 0.5% streptomycin (Gibco). Two million cells were plated in 10 cm tissue culture plates and co-transfected the next day with 475 ng of plasmid encoding bxb1 recombinase and 7125 ng of the plasmid library using Lipofectamine 3000 (Invitrogen). Cells were grown at 33°C for 4 days after transfection. Two days after transfection, the cells were induced with 2 μg/mL doxycycline. All cells were expanded into 15 cm tissue culture plates 6 days after transfection, and 10 million cells were re-plated into new 15 cm tissue culture plates 3 days later.

Twelve days after transfection, the cells were washed with 1× phosphate-buffered saline (PBS, Gibco) and harvested with TrypLE Express protease (Gibco). Cells were washed twice with 2% fetal bovine serum in PBS (wash buffer), resuspended in PBS containing 2% FBS to 10 M cells/mL, and then sorted on a BD FACS Aria II (BD Biosciences, Franklin Lakes, NJ). Recombined cells were isolated based on their characteristic gain of bicistronic eGFP and loss of BFP expression (*Matreyek et al., 2017*). Forward and side scatter profiles were first used to gate for intact live cells, and cells with eGFP-positive (488 nm laser, 530/30 nm emission filter) and BFP-negative (405 nm laser, 450/50 nm emission filter) fluorescence profiles were collected in complete media supplemented with 10% FBS (Gibco) and then plated in 10 cm tissue culture plates. Cells were induced with 2 μg/mL doxycycline 24 hr after sorting. Four days after sorting, 10 million cells were passaged into 15 cm tissue culture plates.

Seven days after sorting for eGFP-positive cells, cells were sorted again according to the surface expression of GnRHR variants. Briefly, cells were first washed with 1× PBS (Gibco) and harvested with TrypLE Express protease (Gibco). GnRHRs expressed at the cell surface were immunostained for 30 min in the dark with DyLight 550-conjugated anti-HA antibody (Invitrogen). Cells were then washed twice with PBS containing 2% FBS, and sorted on a BD FACS Aria II (BD Biosciences). Forward and side scatter profiles were used to gate for intact live cells, and eGFP fluorescence was used to gate

for recombined cells. Cells were then sorted into even quartiles based on the amount of DyLight 550 fluorescence (561 nm laser, 585/15 nm emission filter). Sorted cells were collected in complete media supplemented with 10% FBS (Gibco) and plated in 10 cm tissue culture plates. Cells were allowed to grow to confluency and then were washed with 1 X PBS (Gibco) and harvested with TrypLE Express protease (Gibco). Cell pellets were frozen for subsequent analysis.

## Extraction of genomic DNA and next-generation sequencing

Genomic DNA (gDNA) was extracted from cell pellets using the Sigma GenElute Mammalian gDNA Miniprep Kit (Sigma-Aldrich). DNA amplicons for Illumina sequencing were produced from the gDNA using a previously described semi-nested polymerase chain reaction (PCR) technique (*Matreyek et al., 2017*). In order to selectively amplify recombined DNA, the first PCR utilized a primer which anneals upstream of the 10N UMI and a primer that anneals in the BFP-encoding region of the landing pad. Eight replicate PCRs were carried out with 2.5 µg of gDNA and HiFi HotStart Ready Mix (Kapa Biosystems, Wilmington, MA). These PCRs were limited to seven cycles to avoid PCR bias. PCR products were purified using a ZR-96 DNA Clean & Concentrator-5 kit (Zymo Research), and replicate products were then pooled. 10 µL of this first PCR product was then used as the template for a second PCR, which utilized primers that incorporate Illumina adapter sequences to both ends of the 10N UMI. To avoid PCR bias, these reactions were monitored by real-time PCR on a StepOne RT-PCR instrument (Applied Biosystems, Waltham, MA) and terminated during mid-log amplification (17 or 18 cycles). Four replicate PCRs were carried out with HiFi HotStart Ready Mix (Kapa Biosystems) and SYBR Green fluorescent dye (Applied Biosystems). Replicate PCR products were then combined and purified with a Zymoclean Gel DNA Recovery Kit (Zymo Research). To further improve the yield and quality of the amplicons, an additional six-cycle PCR was carried out with the HiFi HotStart Ready Mix (Kapa Biosystems) and primers that preserve the DNA sequence. These PCR products were then purified using a Zymo DNA Clean & Concentrator-5 kit (Zymo Research). Finally, amplicon yield and quality were assessed using an Agilent 2200 TapeStation with D1000 tape (Agilent Technologies, Santa Clara, CA) and a Synergy Neo2 microplate reader (BioTek). Amplicons were sequenced using an Illumina NextSeq150 flow cell, paired end, with 50% PhiX content.

## Plasma membrane expression and epistasis measurements

Illumina sequencing of the recombined UMIs within each sorted cell fraction were matched to the corresponding GnRHR mutants using the PacBio sequencing data and subjected to previously described quality filtering (*Penn et al., 2020*). Estimations of PME for each mutant were then calculated as previously described (*Penn et al., 2020*). PME measurements were averaged across two biological replicates. Within each library, the average fluorescence intensity values of the mutants were ranked and converted into a percentile. Mutants that differed in their percentile values by more than 25% across the two replicates were excluded from analysis. The use of percentile-based filtering overcomes intrinsic limitations associated with the distinct dynamic range constraints that occur because of the distinct surface immunostaining profiles of these three cellular libraries (*Figure 3*). Intensity values that meet these criteria are highly correlated across the two biological replicates (average Spearman's $\rho$ = 0.936 across the three libraries, *Figure 4—figure supplement 1*). To facilitate the comparison of epistatic interactions within a common set of secondary mutations, we excluded mutants that did not pass quality filtering in all three libraries from our analysis. After data filtering, we obtained PME measurements for 251 mutants in all three libraries (*Supplementary file 4*). These PME measurements were then utilized to calculate epistasis scores for the double mutants in the V276T and W107A libraries using the following equation:

$$\varepsilon = ln\left(\omega_{ab}\right) - ln\left(\omega_a\right) - ln\left(\omega_b\right)$$

where $\varepsilon$ is the epistasis score, $\omega_{ab}$ is the fluorescence intensity of the double mutant relative to WT mGnRHR, $\omega_a$ is the fluorescence intensity of the single mutant in the WT library relative to WT mGnRHR, and $\omega_b$ is the fluorescence intensity of the background mutation (either V276T or W107A) relative to WT mGnRHR, as previously described (*Olson et al., 2014*).

## Expression measurements of individual GnRHR variants

HEK283T cells were obtained from ATCC and grown under the same conditions described above. GnRHR variants were transiently expressed in these cells, and the plasma membrane and intracellular expression of these variants was measured by flow cytometry, as previously described (*Chamness et al., 2021*). The reported expression levels represent measurements from three biological replicates.

## In vitro translation of chimeric Lep proteins

Messenger RNA (mRNA) of chimeric Lep proteins was generated and used for in vitro translation reactions as previously described (*Chamness et al., 2021*). Translation products were then analyzed by SDS-PAGE as previously described (*Chamness et al., 2021*). The intensities of singly (*G1*) and doubly (*G2*) glycosylated protein were quantified by densitometry in ImageJ software. Apparent transfer free energy values representing transfer of the TMDs of interest from the translocon to the membrane were then calculated using the following equation:

$$\Delta G_{app} = -RTln\left(K_{app}\right) = -RTln\left(\frac{G1}{G2}\right)$$

where $\Delta G_{app}$ is the free energy for transfer of the TMD into the membrane, $R$ is the universal gas constant, $T$ is the temperature, $K_{app}$ is the equilibrium constant for transfer of the TMD into the membrane, $G1$ is the intensity of the singly glycosylated band, and $G2$ is the intensity of the doubly glycosylated band, as previously described (*Hessa et al., 2005*). The reported transfer free energy values represent the average of three experimental replicates.

## Structural modeling of *Mus musculus* GnRHR

The sequence of mGnRHR was obtained from UniProt (accession number Q01776) and a homology model in the inactive state was generated as previously described (*Chamness et al., 2021*). Images were generated by structurally aligning this mGnRHR model to an experimentally determined inactive-state human GnRHR crystal structure (GNRHR, PDB 7BR3, 2.79 Å) using the Super command in PyMol (Schrödinger, Inc, New York, NY) (*Yan et al., 2020*).

## Computational estimates for the stability of mGnRHR variants

Structural estimates for the effects of mutations on the stability of mGnRHR were determined using a previously described Rosetta based protocol featuring a membrane protein optimized energy function (*Alford et al., 2015*; *Barlow et al., 2018*). Briefly, the homology model of mGnRHR described above was used as the starting structure for computational stability estimates. A spanfile describing the transmembrane regions was created for mGnRHR using OCTOPUS predictions from Topcons (*Bernsel et al., 2009*; *Viklund and Elofsson, 2008*). The homology model was transformed into membrane coordinates using the mp_transform application. An ensemble of structures of each variant and wildtype (50 iterations) was generated from the homology model. For variants containing two mutations, the mutations were both introduced at the same time. The ΔΔG for each variant was calculated by subtracting the average score of the three lowest scoring variant models from the average score of the three lowest scoring WT models (ΔΔG = ΔG$_{mut}$ - ΔG$_{wt}$).

### Unsupervised learning

To classify mGnRHR variants based on their expression profiles, we first standardized their scores using the Standard Scaler tool from Python's scikit-learn library in order to ensure that the clustering algorithms are not affected by the scale of the data. Next, we then reduced the dimensionality of the standardized data using UMAP (*McInnes et al., 2018*). We then used a density-based clustering algorithm (HDBSCAN) to cluster variants based on their expression profiles (*McInnes et al., 2017*). Unsupervised learning analyses were carried out based on deep mutational scanning measurements alone and did not incorporate any structural and/ or energetic parameters.

### Quantification and analysis

Flow cytometry data were analyzed in FlowJo software (Treestar, Ashland, OR), and in vitro translation data were analyzed in ImageJ software. PME measurements were analyzed in Origin

software (OriginLab, Northampton, MA) and mapped onto the mGnRHR homology model in PyMol (Schrödinger, Inc).

## Acknowledgements

We thank Christiane Hassel and the Indiana University Bloomington Flow Cytometry Core Facility, as well as Luke Tallon, Xuechu Zhao, Lisa Sadzewicz, and the University of Maryland Institute for Genome Sciences for their experimental support. This work was supported by a National Science Foundation (NSF) Graduate Research Fellowship (1342962 to LMC) and by the National Institutes of Health (R01GM129261 to JPS). Any opinions, findings, and conclusions or recommendations expressed in this material are those of the author(s) and do not necessarily reflect the views of the NSF.

## Additional information

### Funding

| Funder | Grant reference number | Author |
|---|---|---|
| National Science Foundation | 1342962 | Laura M Chamness |
| National Institutes of Health | R01GM129261 | Jonathan P Schlebach |

The funders had no role in study design, data collection and interpretation, or the decision to submit the work for publication.

### Author contributions

Laura M Chamness, Conceptualization, Data curation, Formal analysis, Funding acquisition, Validation, Investigation, Visualization, Methodology, Writing - original draft, Writing - review and editing; Charles P Kuntz, Conceptualization, Data curation, Software, Formal analysis, Supervision, Validation, Investigation, Visualization, Methodology, Project administration, Writing - review and editing; Andrew G McKee, Data curation, Validation, Investigation, Methodology, Writing - review and editing; Wesley D Penn, Conceptualization, Formal analysis, Methodology, Project administration, Writing - review and editing; Christopher M Hemmerich, Data curation, Software, Formal analysis, Methodology, Writing - review and editing; Douglas B Rusch, Data curation, Formal analysis, Validation, Methodology, Project administration, Writing - review and editing; Hope Woods, Data curation, Software, Formal analysis, Investigation, Visualization, Methodology, Writing - review and editing; Dyotima, Investigation, Methodology, Writing - review and editing; Jens Meiler, Conceptualization, Software, Supervision, Funding acquisition, Project administration, Writing - review and editing; Jonathan P Schlebach, Conceptualization, Resources, Data curation, Formal analysis, Supervision, Funding acquisition, Validation, Investigation, Visualization, Methodology, Writing - original draft, Project administration, Writing - review and editing

### Author ORCIDs

Laura M Chamness ![ORCID] https://orcid.org/0000-0002-3971-2799
Andrew G McKee ![ORCID] https://orcid.org/0000-0003-1238-987X
Hope Woods ![ORCID] https://orcid.org/0000-0001-6615-545X
Jonathan P Schlebach ![ORCID] https://orcid.org/0000-0003-0955-7633

Reviewer #1 (Public Review): https://doi.org/10.7554/eLife.92406.3.sa1
Reviewer #2 (Public Review): https://doi.org/10.7554/eLife.92406.3.sa2
Author response https://doi.org/10.7554/eLife.92406.3.sa3

## Additional files

### Supplementary files

- MDAR checklist
- Supplementary file 1. Predicted and measured apparent transfer free energies of mGnRHR TM6.
- Supplementary file 2. Solvent accessible surface area of mGnRHR residues.
- Supplementary file 3. mGnRHR library mutants with associated oligo sequences.
- Supplementary file 4. Deep mutational scanning measurements, epistasis scores, and cluster identities.

### Data availability

Code for the analysis of sequencing data for deep mutational scanning experiments can be found on the Schlebach Lab GitHub page (copy archived at *Chamness et al., 2024*). Illumina and PacBio sequencing data is freely available through NCBI. Fluorescence intensity values for mutants in all three mutational libraries are included as Excel files in the Supplementary Materials.

The following dataset was generated:

| Author(s) | Year | Dataset title | Dataset URL | Database and Identifier |
|---|---|---|---|---|
| Charmness LR, Kuntz CP, McKee AG, Penn WD, Hemmerich CM, Rusch DB, Woods H, Meiler J, Schelbach JP | 2024 | Divergent Folding-Mediated Epistasis in the Context of Unstable Membrane Protein Variants | https://www.ncbi.nlm.nih.gov/bioproject/PRJNA1062836 | NCBI BioProject, PRJNA1062836 |

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
