## [Editor Report · eLife assessment]

This **important** study describes exhaustive deep mutational scanning (DMS) of the gonadotropin-releasing hormone wild-type receptor and for two single point mutations that impact its folding and structure, monitoring how plasma membrane expression levels are influenced by the introduced mutations. With **solid** evidence, the authors have pioneered an exploration of the interaction between mutations (epistasis) in a membrane protein, with a potential for explaining membrane protein evolution and genetic diseases.

---

## [Referee Report · Reviewer #1 (Public Review)]

Summary:

The paper carries out an impressive and exhaustive non-sense mutagenesis using deep mutational scanning (DMS) of the gonadotropin-releasing hormone receptor for the WT protein and two single point mutations that (i) influences TM insertion (V267T) and (ii) influences protein stability (W107A) and then measures the effect of these mutants on correct plasma membrane expression (PME).

Overall, most mutations decreased mGnRHR PME levels in all three backgrounds, indicating poor mutational tolerance under these conditions. The W107A variant wasn't really recoverable with low levels of plasma membrane localisation. For the V267T variant, most additional mutations were more deleterious than WT based on correct trafficking, indicating a synergistic effect. As one might expect, there was a higher degree of positive correlation between V267T/W107A mutants and other mutants located in TM regions, confirming that improper trafficking was a likely consequence of membrane protein co-translational folding. Nevertheless, context is important, as positive synergistic mutants in the V27T could be negative in the W107A background and vice versa. Taken together, this important study highlights the complexity of membrane protein folding in dissecting the mechanism-dependent impact of disease-causing mutations related to improper trafficking.

Strengths:

This is a novel and exhaustive approach to dissect how receptor mutations under different mutational backgrounds related to co-translational folding, could influence membrane protein trafficking.

Weaknesses:

The premise for the study requires an in-depth understanding of how the single point mutations analysed effect membrane protein folding in context of DMS, but the single point mutants used could do with further validation. The V267T mutant only reduced MP insertion by 10% and the effect of W107A on protein stability was not assessed. Furthermore, plasma membrane expression has been used as a proxy for incorrect membrane protein folding, but this not necessarily be the case, as even correctly folded membrane proteins may not be trafficked correctly, at least, under heterologous expression conditions. In addition, mutations can effect trafficking and potential post-translational modifications, like glycosylation.

---

## [Referee Report · Reviewer #2 (Public Review)]

Summary:

In this paper, Chamness and colleagues make a pioneering effort to map epistatic interactions among mutations in a membrane protein. They introduce thousands of mutations to the mouse GnRH Receptor (GnRHR), either under wild-type background or two mutant backgrounds, representing mutations that destabilize GnRHR by distinct mechanisms. The first mutant background is W107A, destabilizing the tertiary fold, and the second, V276T, perturbing the efficiency of cotranslational insertion of TM6 to the membrane, which is essential for proper folding. They then measure surface expression of these three mutant libraries, using it as a proxy for protein stability, since misfolded proteins do not typically make it to the plasma membrane. The resulting dataset is then used to shed light on how diverse mutations interact epistatically with the two genetic background mutations. Their main conclusion is that epistatic interactions vary depending on the degree of destabilization and the mechanism through which they perturb the protein. The mutation V276T forms primarily negative (aggravating) epistatic interactions with many mutations, as is common to destabilizing mutations in soluble proteins. Surprisingly, W107A forms many positive (alleviating) epistatic interactions with other mutations. They further show that the locations of secondary mutations correlate with the types of epistatic interactions they form with the above two mutants.

Strengths:

Such a high throughput study for epistasis in membrane proteins is pioneering, and the results are indeed illuminating. Examples of interesting findings are that: (1) No single mutation can dramatically rescue the destabilization introduced by W107A. (2) Epistasis with a secondary mutation is strongly influenced by the degree of destabilization introduced by the primary mutation. (3) Misfolding caused by mis-insertion tends to be aggravated by further mutations. The discussion of how protein folding energetics affects epistasis (Fig. 7) makes a lot of sense and lays out an interesting biophysical framework for the findings.

Weaknesses:

The major weakness comes from the potential limitations in the measurements of surface expression of severely misfolded mutants. It seems that only about 5% of the W107A makes it to the plasma membrane compared to wild-type. This point is discussed quite fairly in the paper. (Figures 2 and 3). This might be a low starting point from which to accurately measure the effects of secondary mutations. I am concerned about the extent to which surface expression can report on protein stability, especially when it comes to double mutants where each mutation alone severely decreases surface expression. It is possible that in these cases, both the single and double mutants are completely misfolded, beyond repair. The surface-expressed proteins in such mutants may not be stable, folded or active at all, and the authors do not provide any indication that the combined effects of the mutations are derived from effects on folding stability or misfolding. Therefore, the reason for the epistatic effects of these mutations is hard to interpret, leaving a notable gap in our understanding. However, I find that this point is discussed much more fairly in the current manuscript.

With that said, I believe that the results regarding the epistasis of V276T with other mutations are strong and very interesting on their own.

Another concern relates to the measurements of the epistatic effects of mutations in the background of the V107A mutation. I am concerned about their measurement accuracy. Firstly, the authors note that the surface immunostaining measurements of these mutants are on average only 2-fold above background, which is quite a low signal-to-noise regimen. Secondly, I believe that the authors still haven't demonstrated the reproducibility of their surface expression measurements. To showcase the reproducibility, the authors show the correlation of two biological replicates in Figure S3. However, these are shown only for the 251 mutations that passed a reproducibility filter, after the authors "discarded variant scores for which the difference in percentile rank across the two replicates was greater than 25%. " . this means that all mutations that showed irreproducible results were filtered out before the analysis in Figure S3. It is, therefore, no surprise that the remaining mutations are highly reproducible, and such an analysis cannot serve as an indication of the reproducibility. It remains possible that a large fraction of the surface immunostaining scores of the V107A variants are dominated by noise and that their correlation in these two replicates might be random and may not necessarily be reproduced in a third replicate, for example.

---

## [Author Response]

The following is the authors’ response to the original reviews.

We are grateful for these balanced, nuanced evaluations of our work concerning the observed epistatic trends and our interpretations of their mechanistic origins. Overall, we think the reviewers have done an excellent job at recognizing the novel aspects of our findings while also discussing the caveats associated with our interpretations of the biophysical effects of these mutations. We believe it is important to consider both of these aspects of our work in order to appreciate these advances and what sorts of pertinent questions remain.

Notably, both reviewers are concerned that our lack of experimental approaches to compare the conformational properties of GnRHR variants weakens our claims. We would first humbly suggest that this constitutes a more general caveat that applies to nearly all investigations of the cellular misfolding of α-helical membrane proteins. Whether or not any current in vitro folding measurements report on conformational transitions that are relevant to cellular protein misfolding reactions remains an active area of debate (discussed further below). Nevertheless, while we concede that our structural and/ or computational evaluations of various mutagenic effects remain speculative, prevailing knowledge on the mechanisms of membrane protein folding suggest our mutations of interest (V276T and W107A) are highly unlikely to promote misfolding in precisely the same way. Thus, regardless of whether or not we were able experimentally compare the relevant folding energetics of GnRHR variants, we are confident that the distinct epistatic interactions formed by these mutations reflect variations in the misfolding mechanism and that they are distinct from the interactions that are observed in the context of stable proteins. In the following, we provide detailed considerations concerning these caveats in relation to the reviewers’ specific comments.

**Reviewer #1 (Public Review):**
The paper carries out an impressive and exhaustive non-sense mutagenesis using deep mutational scanning (DMS) of the gonadotropin-releasing hormone receptor for the WT protein and two single point mutations that (I) influence TM insertion (V267T) and (ii) influence protein stability (W107A), and then measures the effect of these mutants on correct plasma membrane expression (PME).Overall, most mutations decreased mGnRHR PME levels in all three backgrounds, indicating poor mutational tolerance under these conditions. The W107A variant wasn't really recoverable with low levels of plasma membrane localisation. For the V267T variant, most additional mutations were more deleterious than WT based on correct trafficking, indicating a synergistic effect. As one might expect, there was a higher degree of positive correlation between V267T/W107A mutants and other mutants located in TM regions, confirming that improper trafficking was a likely consequence of membrane protein co-translational folding. Nevertheless, context is important, as positive synergistic mutants in the V27T could be negative in the W107A background and vice versa. Taken together, this important study highlights the complexity of membrane protein folding in dissecting the mechanism-dependent impact of disease-causing mutations related to improper trafficking.StrengthsThis is a novel and exhaustive approach to dissecting how receptor mutations under different mutational backgrounds related to co-translational folding, could influence membrane protein trafficking.WeaknessesThe premise for the study requires an in-depth understanding of how the single-point mutations analysed affect membrane protein folding, but the single-point mutants used seem to lack proper validation.

Given our limited understanding of the structural properties of misfolded membrane proteins, it is unclear whether the relevant conformational effects of these mutations can be unambiguously validated using current biochemical and/ or biophysical folding assays. X-ray crystallography, cryo-EM, and NMR spectroscopy measurements have demonstrated that many purified GPCRs retain native-like structural ensembles within certain detergent micelles, bicelles, and/ or nanodiscs. However, helical membrane protein folding measurements typically require titration with denaturing detergents to promote the formation of a denatured state ensemble (DSE), which will invariably retain considerable secondary structure. Given that the solvation provided by mixed micelles is clearly distinct from that of native membranes, it remains unclear whether these DSEs represent a reasonable proxy for the misfolded conformations recognized by cellular quality control (QC, see https://doi.org/10.1021/acs.chemrev.8b00532). Thus, the use and interpretation of these systems for such purposes remains contentious in the membrane protein folding community. In addition to this theoretical issue, we are unaware of any instances in which GPCRs have been found to undergo reversible denaturation in vitro- a practical requirement for equilibrium folding measurements (https://doi.org/10.1146/annurev-biophys-051013-022926). We note that, while the resistance of GPCRs to aggregation, proteolysis, and/ or mechanical unfolding have also been probed in micelles, it is again unclear whether the associated thermal, kinetic, and/ or mechanical stability should necessarily correspond to their resistance to cotranslational and/ or posttranslational misfolding. Thus, even if we had attempted to validate the computational folding predictions employed herein, we suspect that any resulting correlations with cellular expression may have justifiably been viewed by many as circumstantial. Simply put, we know very little about the non-native conformations are generally involved in the cellular misfolding of α-helical membrane proteins, much less how to measure their relative abundance. From a philosophical standpoint, we prefer to let cells tell us what sorts of broken protein variants are degraded by their QC systems, then do our best to surmise what this tells us about the relevant properties of cellular DSEs.

Despite this fundamental caveat, we believe that the chosen mutations and our interpretation of their relevant conformational effects are reasonably well-informed by current modeling tools and by prevailing knowledge on the physicochemical drivers of membrane protein folding and misfolding. Specifically, the mechanistic constraints of translocon-mediated membrane integration provide an understanding of the types of mutations that are likely to disrupt cotranslational folding. Though we are still learning about the protein complexes that mediate membrane translocation (https://doi.org/10.1038/s41586-022-05336-2), it is known that this underlying process is fundamentally driven by the membrane depth-dependent amino acid transfer free energies (https://doi.org/10.1146/annurev.biophys.37.032807.125904). This energetic consideration suggests introducing polar side chains near the center of a nascent TMDs should almost invariably reduce the efficiency of topogenesis. To confirm this in the context of TMD6 specifically, we utilized a well-established biochemical reporter system to confirm that V276T attenuates its translocon-mediated membrane integration (Fig. S1)- at least in the context of a chimeric protein. We also constructed a glycosylation-based topology reporter for full-length GnRHR, but ultimately found its’ in vitro expression to be insufficient to detect changes in the nascent topological ensemble.

In contrast to V276T, the W107A mutation is predicted to preserve the native topological energetics of GnRHR due to its position within a soluble loop region. W107A is also unlike V276T in that it clearly disrupts tertiary interactions that stabilize the native structure. This mutation should preclude the formation of a structurally conserved hydrogen bonding network that has been observed in the context of at least 25 native GPCR structures (https://doi.org/10.7554/eLife.5489). However, without a relevant folding assay, the extent to which this network stabilizes the native GnRHR fold in cellular membranes remains unclear. Overall, we admit that these limitations have prevented us from measuring how much V276T alters the efficiency of GnRHR topogenesis, how much the W107A destabilizes the native fold, or vice versa. Nevertheless, given these design principles and the fact that both reduce the plasma membrane expression of GnRHR, as expected, we are highly confident that the structural defects generated by these mutations do, in fact, promote misfolding in their own ways. We also concede that the degree to which these mutagenic perturbations are indeed selective for specific folding processes is somewhat uncertain. However, it seems exceedingly unlikely that these mutations should disrupt topogenesis and/ or the folding of the native topomer to the exact same extent. From our perspective, this is the most important consideration with respect to the validity of the conclusions we have made in this manuscript.

Furthermore, plasma membrane expression has been used as a proxy for incorrect membrane protein folding, but this not necessarily be the case, as even correctly folded membrane proteins may not be trafficked correctly, at least, under heterologous expression conditions. In addition, mutations can affect trafficking and potential post-translational modifications, like glycosylation.

While the reviewer is correct that the sorting of folded proteins within the secretory pathway is generally inefficient, it is also true that the maturation of nascent proteins within the ER generally bottlenecks the plasma membrane expression of most α-helical membrane proteins. Our group and several others have demonstrated that the efficiency of ER export generally appears to scale with the propensity of membrane proteins to achieve their correct topology and/ or to achieve their native fold (see https://doi.org/10.1021/jacs.5b03743 and https://doi.org/10.1021/jacs.8b08243). Notably, these investigations all involved proteins that contain native glycosylation and various other post-translational modification sites. While we cannot rule out that certain specific combinations of mutations may alter expression through their perturbation of post-translational GnRHR modifications, we feel confident that the general trends we have observed across hundreds of variants predominantly reflect changes in folding and cellular QC. This interpretation is supported by the relationship between observed trends in variant expression and Rosetta-based stability calculations, which we identified using unbiased unsupervised machine learning approaches (compare Figs. 6B & 6D).

**Reviewer #2 (Public Review):**
Summary:In this paper, Chamness and colleagues make a pioneering effort to map epistatic interactions among mutations in a membrane protein. They introduce thousands of mutations to the mouse GnRH Receptor (GnRHR), either under wild-type background or two mutant backgrounds, representing mutations that destabilize GnRHR by distinct mechanisms. The first mutant background is W107A, destabilizing the tertiary fold, and the second, V276T, perturbing the efficiency of cotranslational insertion of TM6 to the membrane, which is essential for proper folding. They then measure the surface expression of these three mutant libraries, using it as a proxy for protein stability, since misfolded proteins do not typically make it to the plasma membrane. The resulting dataset is then used to shed light on how diverse mutations interact epistatically with the two genetic background mutations. Their main conclusion is that epistatic interactions vary depending on the degree of destabilization and the mechanism through which they perturb the protein. The mutation V276T forms primarily negative (aggravating) epistatic interactions with many mutations, as is common to destabilizing mutations in soluble proteins. Surprisingly, W107A forms many positive (alleviating) epistatic interactions with other mutations. They further show that the locations of secondary mutations correlate with the types of epistatic interactions they form with the above two mutants.Strengths:Such a high throughput study for epistasis in membrane proteins is pioneering, and the results are indeed illuminating. Examples of interesting findings are that: (1) No single mutation can dramatically rescue the destabilization introduced by W107A. (2) Epistasis with a secondary mutation is strongly influenced by the degree of destabilization introduced by the primary mutation. (3) Misfolding caused by mis-insertion tends to be aggravated by further mutations. The discussion of how protein folding energetics affects epistasis (Fig. 7) makes a lot of sense and lays out an interesting biophysical framework for the findings.Weaknesses:The major weakness comes from the potential limitations in the measurements of surface expression of severely misfolded mutants. This point is discussed quite fairly in the paper, in statements like "the W107A variant already exhibits marginal surface immunostaining" and many others. It seems that only about 5% of the W107A makes it to the plasma membrane compared to wild-type (Figures 2 and 3). This might be a low starting point from which to accurately measure the effects of secondary mutations.

The reviewer raises an excellent point that we considered at length during the analysis of these data and the preparation of the manuscript. Though we remain confident in the integrity of these measurements and the corresponding analyses, we now realize this aspect of the data required further discussion and documentation which we have provided in the revised version of the manuscript as is described in the following.

Still, the authors claim that measurements of W107A double mutants "still contain cellular subpopulations with surface immunostaining intensities that are well above or below that of the W107A single mutant, which suggests that this fluorescence signal is sensitive enough to detect subtle differences in the PME of these variants". I was not entirely convinced that this was true.

We made this statement based on the simple observation that the surface immunostaining intensities across the population of recombinant cells expressing the library of W107A double mutants was consistently broader than that of recombinant cells expressing W107A GnRHR alone (see Author response image 1 for reference). Given that the recombinant cellular library represents a mix of cells expressing ~1600 individual variants that are each present at low abundance, the pronounced tails within this distribution presumably represent the composite staining of many small cellular subpopulations that express collections of variants that deviate from the expression of W107A to an extent that is significant enough to be visible on a log intensity plot.

Firstly, I think it would be important to test how much noise these measurements have and how much surface immunostaining the W107A mutant displays above the background of cells that do not express the protein at all.

For reference, the average surface immunostaining intensity of HEK293T cells transiently expressing W107A GnRHR was 2.2-fold higher than that of the IRES-eGFP negative, untransfected cells within the same sample- the WT immunostaining intensity was 9.5-fold over background by comparison. Similarly, recombinant HEK293T cells expressing the W107A double mutant library had an average surface immunostaining intensity that was 2.6-fold over background across the two DMS trials. Thus, while the surface immunostaining of this variant is certainly diminished, we were still able to reliably detect W107A at the plasma membrane even under distinct expression regimes. We have included these and other signal-to-noise metrics for each experiment in the Results section of the revised manuscript.

Beyond considerations related to intensity, we also previously noticed the relative intensity values for W107A double mutants exhibited considerable precision across our two biological replicates. If signal were too poor to detect changes in variant expression, we would have expected a plot of the intensity values across these two replicates to form a scatter. Instead, we found DMS intensity values for individual variants to be highly correlated from one replicate to the next (Pearson’s R2 = 0.95, see Author response image 2 for reference). This observation empirically demonstrates that this assay consistently differentiated between variants that exhibit slightly enhanced immunostaining from those that have even lower immunostaining than W107A GnRHR. We have included these discussion points in the Results section as well as scatter plots for replicate variant intensities within all three genetic backgrounds in Figure S3 of the revised manuscript.

**Author response image 2. sa3fig2:** 

But more importantly, it is not clear if under this regimen surface expression still reports on stability/protein fitness. It is unknown if the W107A retains any function or folding at all. For example, it is possible that the low amount of surface protein represents misfolded receptors that escaped the ER quality control.

While we believe that such questions are outside the scope of this work, we certainly agree that it is entirely possible that some of these variants bypass QC without achieving their native fold. This topic is quite interesting to us but is quite challenging to assess in the context of GPCRs, which have complex fitness landscapes that involve their propensity to distinguish between different ligands, engage specific components associated with divergent downstream signaling pathways, and navigate between endocytic recycling/ degradation pathways following activation. In light of the inherent complexity of GPCR function, we humbly suggest our choice of a relatively simple property of an otherwise complex protein may be viewed as a virtue rather than a shortcoming. Protein fitness is typically cast as the product of abundance and activity. Rather than measuring an oversimplified, composite fitness metric, we focused on one variable (plasma membrane expression) and its dominant effector (folding). We believe restraining the scope in this manner was key for the elucidation of clear mechanistic insights.

The differential clustering of epistatic mutations (Fig. 6) provides some interesting insights as to the rules that dictate epistasis, but these too are dominated by the magnitude of destabilization caused by one of the mutations. In this case, the secondary mutations that had the most interesting epistasis were exceedingly destabilizing. With this in mind, it is hard to interpret the results that emerge regarding the epistatic interactions of W107A. Furthermore, the most significant positive epistasis is observed when W107A is combined with additional mutations that almost completely abolish surface expression. It is likely that either mutation destabilizes the protein beyond repair. Therefore, what we can learn from the fact that such mutations have positive epistasis is not clear to me. Based on this, I am not sure that another mutation that disrupts the tertiary folding more mildly would not yield different results. With that said, I believe that the results regarding the epistasis of V276T with other mutations are strong and very interesting on their own.

We agree with the reviewer. In light of our results we believe it is virtually certain that the secondary mutations characterized herein would be likely to form distinct epistatic interactions with mutations that are only mildly destabilizing. Indeed, this insight reflects one of the key takeaway messages from this work- stability-mediated epistasis is difficult to generalize because it should depend on the extent to which each mutation changes the stability (ΔΔG) as well as initial stability of the WT/ reference sequence (ΔG, see Figure 7). Frankly, we are not so sure we would have pieced this together as clearly had we not had the fortune (or misfortune?) of including such a destructive mutation like W107A as a point of reference.

Additionally, the study draws general conclusions from the characterization of only two mutations, W107A and V276T. At this point, it is hard to know if other mutations that perturb insertion or tertiary folding would behave similarly. This should be emphasized in the text.

We agree. Our findings suggest different mutations may not behave similarly, which we believe is a key finding of this work. We have emphasized this point in the Discussion section of the revised manuscript as follows:

“These findings suggest the folding-mediated epistasis is likely to vary among different classes of destabilizing mutations in a manner that should also depend on folding efficiency and/ or the mechanism(s) of misfolding in the cell.”

Some statistical aspects of the study could be improved:(1) It would be nice to see the level of reproducibility of the biological replicates in a plot, such as scatter or similar, with correlation values that give a sense of the noise level of the measurements. This should be done before filtering out the inconsistent data.

We thank the reviewer for this suggestion and will include scatters for each genetic background like the one shown above in Figure S3 of the revised version of the manuscript.

(2) The statements "Variants bearing mutations within the C- terminal region (ICL3-TMD6-ECL3-TMD7) fare consistently worse in the V276T background relative to WT (Fig. 4 B & E)." and "In contrast, mutations that are 210 better tolerated in the context of W107A mGnRHR are located 211 throughout the structure but are particularly abundant among residues 212 in the middle of the primary structure that form TMD4, ICL2, and ECL2 213 (Fig. 4 C & F)." are both hard to judge. Inspecting Figures 4B and C does not immediately show these trends, and importantly, a solid statistical test is missing here. In Figures 4E and F the locations of the different loops and TMs are not indicated on the structure, making these statements hard to judge.

We apologize for this oversight and thank the reviewer for pointing this out. We utilized paired Wilcoxon-Signed Rank Tests to evaluate the statistical significance of these observations and modified the description of these findings in the revised version of the results section as follows:

“Variants bearing mutations within the C-terminal regions including ICL3, TMD6, and TMD7 fare consistently worse in the V276T background relative to WT (paired Wilcoxon-Signed Rank Test p-values of 0.0001, 0.02, and 0.005, respectively) (Fig. 4 B & E). Given that V276T perturbs the cotranslational membrane integration of TMD6 (Fig. S1, Table S1), this directional bias potentially suggests that the apparent interactions between these mutations manifest during the late stages of cotranslational folding. In contrast, mutations that are better tolerated in the context of W107A mGnRHR are located throughout the structure but are particularly abundant among residues in the middle of the primary structure that form ICL2, TMD4, and ECL2 (paired Wilcoxon-Signed Rank Test p-values of 0.0005, 0.0001, and 0.004, respectively) (Fig. 4 C & F).”

(3) The following statement lacks a statistical test: "Notably, these 98 variants are enriched with TMD variants (65% TMD) relative to the overall set of 251 variants (45% TMD)." Is this enrichment significant? Further in the same paragraph, the claim that "In contrast to the sparse epistasis that is generally observed between mutations within soluble proteins, these findings suggest a relatively large proportion of random mutations form epistatic interactions in the context of unstable mGnRHR variants". Needs to be backed by relevant data and statistics, or at least a reference.

We thank the reviewer for this reasonable suggestion. In the revised manuscript, we included the results of a paired Wilcoxon-Signed Rank Test that confirms the statistical significance of this observation and modified the Results section to reflect this as follows:

“Notably, these 98 variants are enriched with TMD variants (65% TMD) relative to the overall set of 251 variants (45% TMD, Fisher’s Exact Test p = 0.0019). These findings suggest random mutations form epistatic interactions in the context of unstable mGnRHR variants in a manner that depends on the specific folding defect (V276T vs. W107A) and topological context.”

**Reviewer #1 (Recommendations for the Authors):**
As far as this reviewer is aware, the effect of the V267T variant on MP insertion has not been measured directly; its position corresponds to T277 in TMD6 of human GnRHR that has been measured for TM insertion, but given the clear lack of conservation (threonine vs valine) the mutation in TM6 could potentially have a different impact on the mouse homologue. Please clarify what the predicted delta TM for insertion is between human and mouse GnRHR is? Moreover, I would argue that single TM insertion by tethering to Lep is insufficient to understand MP insertion/folding, as neighbouring TM helices could help to drive TM6 insertion. Has ER microsome experiments for mouse GnRHR also been carried out in the context of neighbouring helices?

We included measurements (and predictions) of the impact of the V276T substitution on the translocon-mediated membrane integration of the mouse TMD6 in the context of a chimeric Lep protein (see Fig. S1 & Table S1). Our results reveal that this substitution decreases the efficiency of TMD6 membrane integration by ~10%. Though imperfect, this prevailing biochemical assay remains popular for a variety of theoretical and technical reasons. Importantly, extensive experimental testing of this system has shown that these measurements report apparent equilibrium constants that are well-described by two-state equilibrium partitioning models (see DOIs 10.1038/nature03216 and 10.1038/nature06387). This observation provides a reasonable rationale to interpret these measurements using energetic models as we have in this work (see Table S1). From a technical perspective, the Lep system is also advantageous due to the fact that this protein is generally well expressed in the context of in vitro translation systems containing native membranes, which generally ensures a consistent signal to noise and dynamic range for membrane integration measurements. Nevertheless, the reviewers are correct that membrane integration efficiencies are likely distinct in the context of the native mGnRHR protein. For these reasons, we attempted to develop a glycosylation-based topology reporter prior to the posting and submission of this manuscript. However, all GnRHR reporters we tested were poorly expressed in vitro and the resulting 35S-labeled proteins only generated faint smears on our phosphorimaging screens that could not be interpreted. For these reasons, we chose to rely the Lep measurements for these investigations.

The lack of a more relevant topological reporter is one of many challenges we faced in our investigations of this unstable, poorly behaved protein. We share the reviewer’s frustrations concerning the speculative aspects of this work. Nevertheless, there is increasing appreciation for the fact that our perspectives on protein biophysics have been skewed by our continuing choice to focus on the relatively small set of model proteins that are compatible with our favored methodologies (doi: 10.1016/j.tibs.2013.05.001). We humbly suggest this work represents an example of how we can gain a deeper understanding of the limits of biochemical systems when we instead choose to study the unsavory bits of cellular proteomes. But this choice requires a willingness to make some reasonable assumptions and to lean on energetic/ structural modeling from time to time. Despite this limitation, we believe there is still tremendous value in this compromise.

What is the experimental evidence the W107A variant affects the protein structure? Has its melting temperature with and without inverse agonist binding for WT vs the W107A variant been measured, for example? Even heat-FSEC of detergent-solubilised membranes would be informative to know how unstable the W107A variant is. If is very unstable in detergent, then it could be that recovery mutants are going to be unlikely as you are already starting with a poor construct showing poor folding/localisation.

We again understand the rationale for this concern, but do not believe that thermal melting measurements are likely to report the same sorts of conformational transitions involved in cellular misfolding. Heating up a protein to the point in which membranes (or micelles) are disrupted and the proteins begin to form insoluble aggregates is a distinct physical process from those that occur during co- and post-translational folding within intact ER membranes at physiological temperatures (discussed further in the Response to the Reviews). Indeed, as the reviewer points out below, there seems to be little evidence that secretion is linked to thermal stability or various other metrics that others have attempted to optimize for the sake of purification and/ or structural characterization. Thus, we believe it would be just as speculative to suggest thermal aggregation represents a relevant metric for the propensity of membrane proteins to fold in the cell. The physical interpretation of membrane protein misfolding reaction remains contentious in our field due to the key fact that the denatured states of helical membrane proteins remain highly structured in a manner that is hard to generalize beyond the fact that the denatured states retain α-helical secondary structure (doi: 10.1146/annurev-biophys-051013-022926). This is in stark contrast to soluble proteins, where random coil reference states have proven to be generally useful for energetic interpretations of protein stability. For reference, our lab is currently working to leverage epistatic measurements like this to map the prevailing physiological denatured states of an integral membrane protein. Our current findings suggest that non-native electrostatic interactions form in the context of misfolded states. We hope that more information on the structural aspects of these states will help us to develop and interpret meaningful folding measurements within the membrane.

For reference, even in cases when quantitative folding measurements can be achieved, their relevance remains actively debated. As a point of reference, the corresponding author of this work previously worked on the stability and misfolding of another human α-helical membrane protein (PMP22). Like GnRHR, PMP22 is prone to misfolding in the secretory pathway and is associated with dozens of pathogenic mutations that cause protein misfolding. To understand how the thermodynamic stability of this protein is linked to secretion, the corresponding author purified PMP22, reconstituted it into n-Dodecyl-phosphocholine (DPC) micelles, and measured its resistance to denaturation by an anionic denaturing detergent (Lauryl Sarcosine, LS). The results were initially perplexing due to the fact that equilibrium unfolding curves manifested as an exponential decay (rather than a sigmoid) and relaxation kinetics appeared to be dominated by the rate constant for unfolding (doi: 10.1021/bi301635f). Unfortunately, these data could not be fit with existing folding models due to the lack of a folded protein baseline and the absence of a folding arm in the chevron plot. We eventually found that a full sigmoidal unfolding transition and refolding kinetics could be measured upon addition of 15% (v/v) glycerol. Our measurements revealed that the free energy of unfolding in DPC micelles was 0 kcal/ mol (without glycerol). This shocking lack of WT stability made it impossible to directly measure the effects of destabilizing mutations that enhance misfolding- you can’t measure the unfolding of a protein that is already unfolded. We ultimately had to instead infer the energetic effects of such mutations from the thermodynamic coupling between cofactor binding and folding (doi: 10.1021/jacs.5b03743). Finally, after demonstrating the resulting ΔΔGs correlated with both cellular trafficking and disease phenotype, we still faced justified scrutiny about the relevance of these measurements due to the fact that they were carried out in micelles. For these reasons, we do not feel that additional biophysical measurements will add much to this work until more is understood about the nature of misfolding reactions in the membrane and how to effectively recapitulate it in vitro. We also note that PMP22 is secreted with 20% efficiency in mammalian cell lines, which is 20-fold more efficient than human GnRHR under similar conditions (doi: 10.1016/j.celrep.2021.110046). Thus, we suspect equilibrium unfolding measurements are likely out of reach using previously described measurements.

Our greatest evidence suggesting W107A destabilizes the protein has to do with the fact that it deletes a highly conserved structural contact and that this structural modification kills its secretion. The fact that this mutation clearly reduces the escape of GnRHR from ER quality control is a classic indicator of misfolding that represents the cell’s way of telling us that the mutation compromises the folding of the nascent protein in some way or another. Precisely how this mutation remodels the nascent conformational ensemble of nascent GnRHR and how this relates to the free energy difference between the native and non-native portions of its conformational ensemble under cellular conditions is a much more challenging question that lies beyond the scope of this investigation (and likely beyond the scope of what’s currently possible). Indeed, there is an entire field dedicated to understanding such. Nevertheless, the difference in the epistatic interactions formed by W107A and V276T is at the very least consistent with our speculative interpretation that these two mutations vary in their misfolding mechanism and/ or in the extent to which they destabilize the protein. For these reasons, we feel the main conclusions of this manuscript are well-justified.

Please clarify if the protein is glycosylated or not and, if it is, how would this requirement affect the conclusions of your analysis?

As we noted in the Response to the Reviewers, which also constitutes a published portion of the final manuscript, this protein is indeed glycosylated. We were well aware of this aspect of the protein since inception of this project and do not think this changes our interpretation at all. Most membrane proteins are glycosylated, and several groups have demonstrated in various ways that the secretion efficiency of glycoproteins is proportional to certain stability metrics for secreted soluble proteins and membrane proteins alike. Generally, mutations that enhance misfolding do not change the propensity of the nascent chain to undergo N-linked glycosylation, which occurs during translation before protein synthesis and/ or folding is complete. Misfolded proteins typically carry lower weight glycans, which reflects their failure to advance from the ER to the Golgi, where N-linked glycans are modified and O-linked glycans are added. From our perspective, glycosyl modifications just ensure that nascent proteins are engaged by calnexin and other lectin chaperones involved in QC. It does not decouple folding from secretion efficiency. In the case of PMP22 (described above), we found that removal of its glycosylation site allows the nascent protein to bypass the lectin chaperones in a manner that enhances its plasma membrane expression eight-fold (doi: 10.1016/j.jbc.2021.100719). Similar to WT, the expression of several misfolded PMP22 variants also significantly increases upon removal of the glycosylation site. Nevertheless, their expression is still significantly lower than the un-glycosylated WT protein, and the expression patterns of the mutants relative to WT was quite similar across this panel of un-glycosylated proteins. Thus, while glycosylation certainly impacts secretion, it does not change its dependence on folding efficiency within the ER. There are many layers of partially redundant QC within the ER, and it seems that folding imposes a key bottleneck to secretion regardless of which QC proteins are involved. For these reasons, we do not think glycosylation (or other PTMs) should factor into our interpretation of these results.

One caveat with the study is that there is a poor understanding of the factors that decide if the protein should be trafficked to the PM or not. Even secretory proteins not going through the calnexin/reticulum cycle (as they have no N-linked glycans), might still get stuck in the ER, despite the fact they are functional. Could this be a technical issue of heterologous expression overloading the Sec system?

While we agree that there is much to be learned about this topic, we disagree with the notion that our understanding of folding and secretion is insufficient to generally interpret the molecular basis of the observed trends. In collaboration with various other groups, the corresponding author of this paper has shown for several other proteins that the stability of the native topology and the native tertiary structure can constrain secretion efficiency (see dois: 10.1021/jacs.8b08243, 10.1021/jacs.5b03743, and 10.1016/j.jbc.2021.100423). Moreover, the Balch and Kelly groups demonstrated many years ago that relatively simple models for the coupling between folding and chaperone binding can recapitulate the observed effects of mutations on the secretion efficiency of various proteins (doi: 10.1016/j.cell.2007.10.025). Given a wide body of prevailing knowledge in this area, we believe it is entirely reasonable to assume that the conformational effects of these mutation have a dominant effect on plasma membrane expression.

Whether or not some of the proteins retained in the ER are folded and/ or functional is an interesting question, but is outside the scope of this work. Various lines of evidence concerning approaches to rescue misfolded membrane proteins suggest many of these variants are likely to retain residual function once they escape the ER, which may suggest there are pockets of foldable/ folded proteins within the ER. But it seems generally clear that the efficiency of folding in the ER bottlenecks secretion regardless of whether or not the ER contains some fraction of folded/ functional protein. We note that it is certainly possible, if not likely, that secretion efficiency is likely to be higher at lower expression levels (doi: 10.1074/jbc.AC120.014940). However, the mutational scanning platform used in this work was designed such that all variants are expressed from an identical promoter at the same location within the genome. Thus, for the purposes of these investigations, we believe it is entirely fair to draw “apples-to-apples” comparisons of their relative effects on plasma membrane expression.

Please see Francis Arnold's paper on this point and their mutagenesis library of the channelrhodopsin (https://www.pnas.org/doi/10.1073/pnas.1700269114), which further found that 20% of mutations improved WT trafficking. Some general comparisons to this paper might be informative.

We agree that it may be interesting to compare the results from this paper to those in our own. Indeed, we find that 20% of the point mutations characterized herein also enhance the expression of WT mGnRHR, as mentioned in the Results section. However, we think it might be a bit premature to suggest this is a more general trend in light of the fact that the channelrhodopsins engineered in those studies were not of eukaryotic origin and have likely resulted from distinct evolutionary constraints. We ultimately decided against adding more on this to our already lengthy discussion in order to maintain focus on the mechanisms of epistasis.

Chris Tate and others have shown that there is a high frequency of finding stabilising point mutations in GPCRs and this is the premise of the StAR technology used to thermostabilise GPCRs in the presence of different ligands, i.e. agonist vs inverse agonists. As far as I am aware, there is a poor correlation between expression levels and thermostability (measured by ligand binding to detergent-solubilised membranes). As such, it is possible that some of the mutants might be more stable than WT even though they have lower levels of PME.

We believe the disconnect between thermostability and expression precisely speaks to our main point about the suitability of current membrane protein folding assays for the questions we address herein. The degradative activity of ER quality control has not necessarily selected for proteins that are resistant to thermal degradation and/ or are suitable for macromolecular crystallography. For this reason, it is often not so difficult to engineer proteins with enhanced thermal stability. We do not believe this disconnect signals that quality control is insensitive to protein folding and stability, but rather that it is more likely to recognize conformational defects that are distinct from those involved in thermal degradation and/ or aggregation. Indeed, recent work from the Fluman group, which builds on a wider body of previous observations, has shown that the exposure of polar groups within the membrane is a key factor that recruits degradation machinery (doi: 0.1101/2023.12.12.571171). It is hard to imagine that these sorts of conformational defects are the same as those involved in thermal aggregation.

**Reviewer #2 (Recommendations For The Authors):**
(1) I believe that by focusing more on the epistasis with V276T, and less on W107A, the paper could be strengthened significantly.

We appreciate this sentiment. But we believe the comparison of these two mutants really drive home the point that destabilizing mutations are not equivalent with respect to the epistatic interactions they form.

(2) In the abstract - please define the term epistasis in a simple way, to make it accessible to a general audience. For example - negative epistasis means that... this should be explicitly explained.

We thank the reviewer for this suggestion. To meet eLife formatting, we had to cut down the abstract significantly. We simplified this as best we could in the following statement:

“Though protein stability is known to shape evolution, it is unclear how cotranslational folding constraints modulate the synergistic, epistatic interactions between mutations.”

We also define positive and negative epistasis in the results section as follows:

“Positive Ɛ values denote double mutants that have greater PME than would be expected based on the effects of single mutants. Negative Ɛ values denote double mutants that have lower PME than would be expected based on the effects of single mutants. Pairs of mutations with Ɛ values near zero have additive effects on PME.”

(3) The title is quite complex and might deter readers from outside the protein evolution field. Consider simplifying it.

We thank the reviewer for this suggestion. We have simplified the title to the following:

“Divergent Folding-Mediated Epistasis Among Unstable Membrane Protein Variants”

(4) The paper could benefit from a simple figure explaining the different stages of membrane protein folding (stages 1+2) to make it more accessible to readers from outside the membrane protein field.

This is a great suggestion. We incorporated a new schematic in the revised manuscript that outlines the nature of these processes (see Fig. 1A in the revised manuscript).

(5) For the FACS-Seq experiment - it was not clear to me if and when all cells are pulled together. For example - are the 3 libraries mixed together already at the point of transfection, or are the transfected cells pulled together at any point before sorting? This could have some implications on batch effects and should, therefore, be explicitly mentioned in the main text.

We thank the reviewer for this suggestion. We modified the description of the DNA library assembly to emphasize that the mutations were generated in the context of three mixed plasmid pools, which were then transfected into the cells and sorted independently:

“We then generated a mixed array of mutagenic oligonucleotides that collectively encode this series of substitutions (Table S3) and used nicking mutagenesis to introduce these mutations into the V276T, W107A, and WT mGnRHR cDNAs (Medina-Cucurella et al., 2019), which produced three mixed plasmid pools.”

(6) The following description in the text is quite confusing. It would be better to simplify it considerably or remove it: "scores (Ɛ) were then determined by taking the log of the double mutant fitness value divided by the difference between the single mutant fitness values (see Methods)."

We thank the reviewer for this valuable feedback and have simplified the text as follows:

“To compare epistatic trends in these libraries, we calculated epistasis scores (Ɛ) for the interactions that these 251 mutations form with V276T and W107A by comparing their relative effects on PME of the WT, V276T, and W107A variants using a previously described epistasis model (product model, see Methods) (Olson et al. 2014).”